# A comprehensive single-cell map of T cell exhaustion-associated immune environments in human breast cancer

Sandra Tietscher[1,2,3], Johanna Wagner [1,8], Tobias Anzeneder[4], Claus Langwieder[5], Martin Rees[5], Bettina Sobottka[6], Natalie de Souza[1,7] & Bernd Bodenmiller [1,2] ✉

Immune checkpoint therapy in breast cancer remains restricted to triple negative patients, and long-term clinical benefit is rare. The primary aim of immune checkpoint blockade is to prevent or reverse exhausted T cell states, but T cell exhaustion in breast tumors is not well understood. Here, we use single-cell transcriptomics combined with imaging mass cytometry to systematically study immune environments of human breast tumors that either do or do not contain exhausted T cells, with a focus on luminal subtypes. We find that the presence of a PD-1$^{high}$ exhaustion-like T cell phenotype is associated with an inflammatory immune environment with a characteristic cytotoxic profile, increased myeloid cell activation, evidence for elevated immunomodulatory, chemotactic, and cytokine signaling, and accumulation of natural killer T cells. Tumors harboring exhausted-like T cells show increased expression of MHC-I on tumor cells and of CXCL13 on T cells, as well as altered spatial organization with more immature rather than mature tertiary lymphoid structures. Our data reveal fundamental differences between immune environments with and without exhausted T cells within luminal breast cancer, and show that expression of PD-1 and CXCL13 on T cells, and MHC-I – but not PD-L1 – on tumor cells are strong distinguishing features between these environments.

Immune checkpoint blockade therapies have improved patient outcomes in many human cancer types[1–4]. Breast cancer is an exception, previously believed to be due to poor immunogenicity[5], but several studies have identified a strong influence of the immune infiltrate on breast cancer progression[6,7]. T cell infiltration in particular impacts patient survival for all breast cancer subtypes, but the effect on prognosis may be positive or negative depending on the subtype[8],

highlighting the complexity of the tumor-immune interaction. Recently, atezolizumab and pembrolizumab, two checkpoint inhibitors targeting the PD-1/PD-L1 pathway, have been approved for patients with triple-negative breast cancer[9]. These treatments increase progression-free survival by several months, but most patients do not gain long-term clinical benefit, and there are no approved immunotherapeutic options for patients with luminal HER2-negative breast

[1]Department of Quantitative Biomedicine, University of Zurich, Zurich, Switzerland. [2]Institute for Molecular Health Sciences, ETH Zurich, Zurich, Switzerland. [3]Life Science Zurich Graduate School, ETH Zurich and University of Zurich, Zurich, Switzerland. [4]Patients' Tumor Bank of Hope (PATH), Munich, Germany. [5]Pathology at Josefshaus, Dortmund, Germany. [6]Department of Pathology and Molecular Pathology, University Hospital Zurich and University of Zurich, Zurich, Switzerland. [7]Institute of Molecular Systems Biology, ETH Zurich, Zurich, Switzerland. [8]Present address: Division of Translational Medical Oncology, German Cancer Research Center (DKFZ) and National Center for Tumor Diseases (NCT) Heidelberg, Heidelberg, Germany. ✉e-mail: bernd.bodenmiller@uzh.ch

cancer subtypes. Our understanding of the mechanisms of resistance or response to immunotherapy is incomplete, as is our knowledge of the complex cellular interactions in the tumor immune micro-environment (TIME). In order to design new immunotherapies and to effectively use existing ones for luminal breast cancer patients, it is critical to understand the TIME as a whole.

T cells make up a large part of the immune infiltrate in most tumors, including breast tumors[8], and provide an important line of defense against cancer cells. Upon activation, cytotoxic T lymphocytes (CTLs) can exert their effector function through cytolytic molecules such as granzymes and granulysin or via inflammatory cytokines such as IFN-γ and TNF-α[10,11]. At sites of chronic inflammation—such as tumors—where the elimination of target cells fails and T cell receptor (TCR) signaling persists, T cells may become "exhausted", a cellular state characterized by an elevated co-expression of inhibitory check-point receptors such as PD-1, CTLA-4, TIM-3, and LAG-3[12]. Prevention or reversal of T cell exhaustion, also termed T cell dysfunction, is the primary aim of most immune checkpoint blockade therapies[12]. Originally discovered under conditions of viral infection, T cell exhaustion is associated with reduced effector functions. Although loss of pro-liferative potential was thought to be a characteristic of exhausted T cells, recent evidence suggests that they may still proliferate in human tumors[13,14] and that T cells with an exhausted phenotype expand after anti-PD1 treatment in breast tumors[15], but the extent to which these cells retain their anti-tumor functions is unclear[16].

Myeloid cells are also present in the immune infiltrate in most breast tumors, with pro- and anti-tumor effects reported[17]. Tumor-associated macrophages (TAMs) can directly suppress T cell responses by expression of PD-L1 and other immunomodulatory molecules, or indirectly via recruitment of regulatory T cells (Tregs). Dendritic cells (DCs) prime anti-tumor T cell function by surface-presentation of tumor-derived antigens[18]. Myeloid cells may also promote angiogen-esis and tissue remodeling[19], and other immune cell types like B cells and natural killer (NK) cells further add to the complexity of immune-mediated effects on tumors.

Recent technological developments allow for increasingly robust and comprehensive single-cell analysis of tumor microenvironments at the transcript and protein levels[20–22], facilitating the investigation of cell type abundances, cellular phenotypes, and cell-cell communica-tion. Spatially resolved multiplexed methods further enable the study of cellular neighborhoods and spatially defined tissue patterns such as tertiary lymphoid structures (TLS) in cancer[23–26]. Although tran-scriptomic studies have provided insight into the immune composi-tion of both treatment naïve and anti-PD1 treated breast tumors at the single cell level[15,27,28], a more systematic analysis of TIMEs with and without evidence of T cell exhaustion is missing. In previous work, we used mass cytometry-based cellular phenotyping to classify luminal breast tumors into three groups based on infiltrating T cell and mye-loid cell phenotypes[29]. One group showed strong enrichment of an exhaustion-like T cell phenotype. This group accounted for around 13% of all tumors, included both estrogen receptor positive (ER)+ and ER− tumors, and might be amenable to immune checkpoint inhibitor therapy. A second group, accounting for 48% of all tumors, also con-tained a high number of T cells but did not display signs of immune exhaustion.

In this work, we use single-cell RNA sequencing (scRNA-seq) combined with imaging mass cytometry (IMC) to systematically ana-lyze the TIME of breast tumors with and without signs of T cell exhaustion (called "exhausted" and "non-exhausted" environments hereafter). We report that T cells expressing exhaustion markers dis-play hallmarks of tumor reactivity and proliferation, and that their presence coincides with elevated major histocompatibility class I (MHC-I) expression on tumor cells. High levels of expression of *GZMB* and *FASL* (encoding Granzyme B and Fas ligand) but low levels of *IFNG* and *TNF* (encoding Interferon-γ and TNF-α) in these exhausted-like T cells are suggestive of an altered cytotoxic profile but remaining capacity for tumor cell killing. CXCL13 is a common marker of PD-1^high CD8+ and PD-1^high CD4+ T cells and we find that sites of B cell and CXCL13^high T cell accumulation, but not mature TLS, are more frequent in exhausted immune environments. Exhausted breast tumor envir-onments are also enriched in cytotoxic CSF-1+ natural killer T (NKT) cells and show evidence of inflammation. Finally, we provide a map of cellular interactions within the breast TIME, predicting strongly ele-vated immunomodulatory, chemotactic, and cytokine signaling in tumors enriched in exhausted-like T cells. Our findings suggest that the fundamental differences between immune environments with and without signs of T cell exhaustion may be explained by different immune escape mechanisms—avoidance of tumor-specific T cell acti-vation in non-exhausted environments and progressive T cell dys-function through chronic inflammatory signaling in exhausted environments. We describe exhaustion-associated TIME character-istics that might facilitate the discovery of new therapeutic targets and propose PD-1, CXCL13 and MHC-I as a new biomarker combination for patient stratification.

## Results
### A single-cell map of immune environments in breast tumors
In order to systematically study the TIME, we performed droplet-based scRNA-seq of 14 immune-infiltrated breast tumors (Fig. 1a and Sup-plementary Data 1), 12 of which were from a cohort we had previously analyzed by mass cytometry[29]. In this previous work, the tumors had been grouped according to shared patterns of tumor and immune cells, and we now selected the tumors based on the previous grouping. Half of the samples contained exhausted T cells (i.e., PD-1^high/CTLA-4^high/CD38^high T cells); we annotated these as immune environment 1 (IE1) and refer to them as exhausted environments. The other half mainly contained T cells that did not express exhaustion markers and were annotated as immune environment 2 (IE2); we refer to these as non-exhausted environments. With one exception, all tumors analyzed by scRNA-seq were classified as luminal; although all tumor grades were represented, most tumors were of grade 3 (Supplemen-tary Fig. 1A).

We subjected live, dissociated cells with no cell type enrichment to single-cell transcriptome sequencing using the 10x Genomics plat-form. After standard data pre-processing, 119,000 high-quality cell measurements remained in the dataset (Supplementary Fig. 1B and Supplementary Data 2), and no batch effect was apparent for indivi-dual tumors (Fig. 1b) or immune environments (Fig. 1c). We manually annotated graph-based clusters (Supplementary Fig. 1C and Supple-mentary Data 3) based on differential expression analysis and known marker genes for the main expected cell types (Fig. 1d, e and Supple-mentary Fig. 1D). This revealed 36,000 T and NK cells, which clustered together due to their transcriptional similarity; 27,000 cells of the myeloid lineage (monocytes, macrophages and DCs, excluding gran-ulocytes); and 18,000 epithelial cells. Other cell types were present at lower frequencies. We did not recover adipocytes and neutrophils, likely because they are particularly vulnerable to sample processing and cryopreservation damage[30,31]. We observed few significant differ-ences in frequencies of the main cell types between tumors with IE1 versus IE2 (Fig. 1f). Importantly, although different T cell phenotypes drove the distinction between IE1 and IE2, the overall T cell frequency was similar in both groups.

To understand our scRNA-seq based findings in the in situ tumor context, we performed IMC on formalin-fixed, paraffin embedded (FFPE) tissue of 12 of the 14 sequenced samples (Fig. 1a). For each sample, two consecutive FFPE sections were stained with two different panels of antibodies and/or mRNA probes (Supplementary Fig. 2A): Panel 1 (the Protein Panel) consists of 42 metal-labeled antibodies, and Panel 2 (the RNA Panel) includes 12 oligonucleotide probes[24], the majority targeting cytokine mRNAs, and 26 antibodies

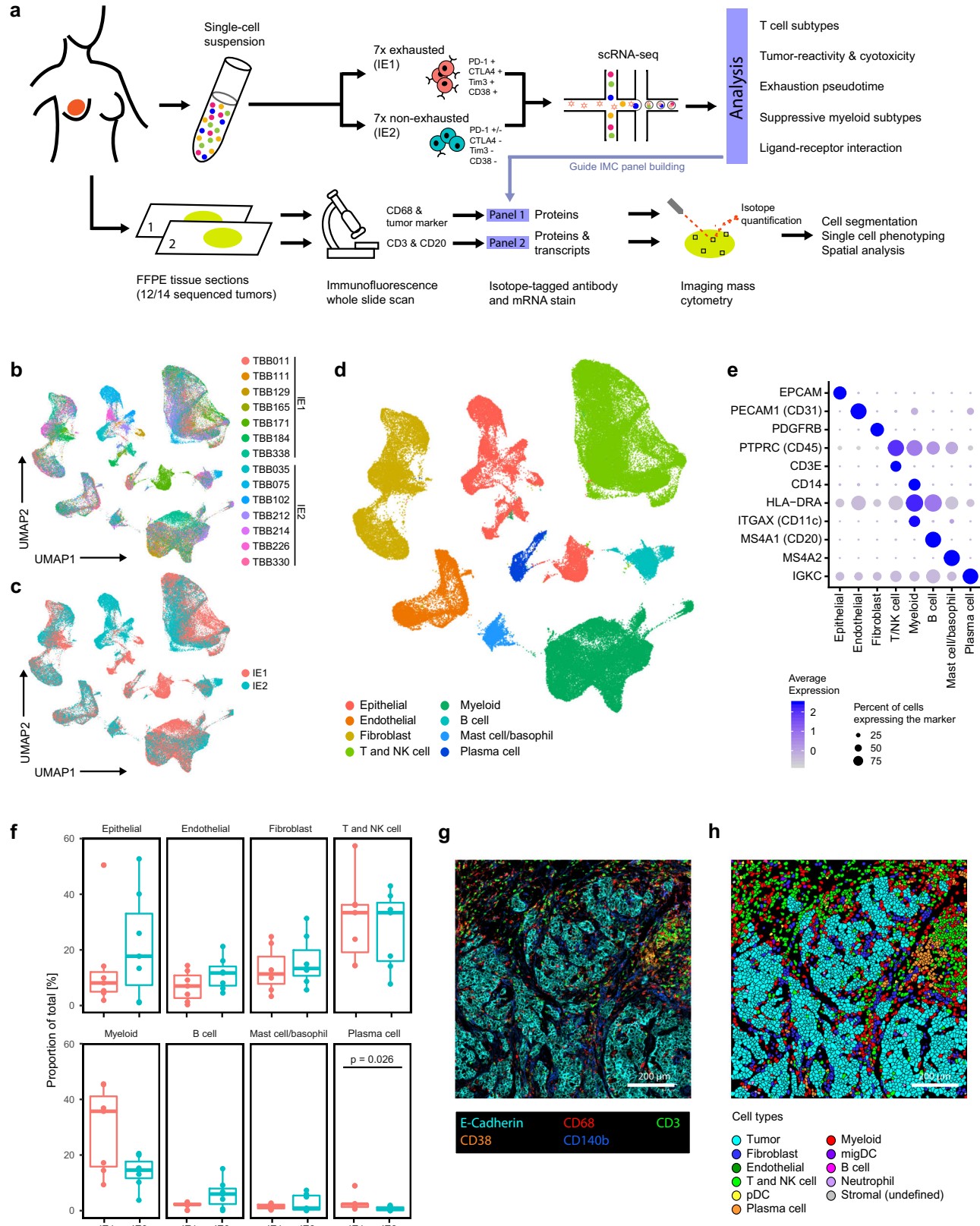

(Supplementary Data 4). Marker choice for both panels was informed by analysis of the scRNA-seq data. Prior to IMC, we used whole-slide immunofluorescence imaging of cell type markers to select four to six regions of interest (ROIs) representative of the general tissue structure. For samples that had immature or mature TLS (defined as sites of B cell accumulation without a clear center or sites with dense round B

cell accumulation, respectively[32]), additional regions including these structures were imaged (Supplementary Fig. 2B). In total, we acquired 77 ROIs for each panel (Fig. 1g and Supplementary Figs. 3 and 4), and data pre-processing and single-cell segmentation resulted in more than 400,000 single cells per panel, with manually matched ROIs for consecutive slices having similar cell numbers (Supplementary

**Fig. 1 | Transcriptomic and spatial proteomic analysis of breast tumor immune environments. a** Sample selection and experimental approach. **b** UMAP plot of scRNA-seq data from all 120,000 cells colored by patient. **c** UMAP plot of scRNA-seq data colored by immune environment (IE). **d** UMAP plot of scRNA-seq data colored by cell type. **e** DotPlot showing transcript expression of main cell type markers in the indicated cell subsets. **f** Proportion (% of total cells) of main cell types in IE1 and IE2 tumors. Cell types were annotated based on marker expression in scRNA-Seq data. Two-sided Wilcoxon rank sum test was used for statistical

analysis. Boxplot centers indicate the group median, boxplot bodies show interquartile ranges (IQR), and whiskers extend to the largest and the smallest value within 1.5 times the IQR above the 75th percentile and below the 25th percentile, respectively. $n = 12$ independent patient samples. **g** Exemplary IMC image showing staining patterns for the indicated markers. **h** Single-cell masks for the IMC image displayed in **g** colored by cell type. IMC staining patterns and single-cell masks were compared for all 77 images with similar results.

Fig. 2C). We performed graph-based clustering based on average marker expression to identify the cell types present (Fig. 1h and Supplementary Figs. 2 and 5). All cell types identified in the scRNA-seq dataset as well as neutrophils were identified in our IMC dataset (Supplementary Fig. 5A); neutrophils were generally rare but were present at slightly higher frequency in IE1 samples than in IE2 samples (Supplementary Fig. 5B). For the RNA Panel IMC dataset, we assigned a binary cytokine expression status to each cell based on the signal intensity for the given cytokine corrected by that of a background mRNA probe complementary to the bacterial *DapB* mRNA[33]. These data were then used for a comprehensive, spatially resolved, immune-focused analysis of the tumor microenvironment.

**PD-1$^{high}$/CTLA-4$^{high}$ T cells are proliferative, express markers of tumor reactivity, and coincide with high MHC-I expression in tumor cells**

The two immune environments in our study cohort differed in protein-level expression of T cell exhaustion markers[29]. To further probe the T cell functional states in these tumors, we analyzed the transcriptomes of T cell subtypes. We also included NK cells in this analysis, since these cells have transcriptional and functional similarities with T cells and can reportedly undergo exhaustion[34].

We first used a pseudobulk approach, averaging read counts across cells for individual patient samples. This identified several genes expressed at higher levels in IE1 than IE2, including transcripts *PDCD1*, *CD276*, and *HAVCR2* encoding immune checkpoint receptors PD-1, B7-H3, and TIM-3, respectively (Fig. 2a), as expected from our previous mass cytometry data[29]. A number of genes normally associated with T cell activation, such as *MKI67* (encoding Ki-67) and *GZMB* (encoding Granzyme B), were also expressed at higher levels in IE1 than IE2 (Fig. 2a). In addition, mRNAs encoding three transcription factors (IRF4, BATF4, and TOX) previously associated with strong and/or chronic TCR signaling[35–37] were expressed at significantly higher levels in T and NK cells from IE1 than from IE2 (Fig. 2b). In contrast, *TCF7*, which encodes a transcription factor positively associated with favorable response to checkpoint therapy in melanoma[38], was elevated in IE2 compared to IE1. Finally, we found that cytokine and cytokine receptor expression was generally higher in T and NK cells from IE1 than from IE2, with the most significant differences seen for *CCL3*, *CXCR6*, *CSF1*, and *IL13*, all of which have roles in inflammatory signaling (Fig. 2c). Running the same analysis while accounting for tumor grade gave similar results (Supplementary Fig. 6).

To compare the phenotypic composition of the T and NK cell compartment in the IE1 and IE2 immune environments and identify which cell types show changes in abundance, we performed graph-based subclustering of the scRNA-seq data and annotated cellular subtypes based on differential expression analysis. This revealed 14 T cell clusters, four NK cell clusters, and one NKT cell cluster (Fig. 2d–f and Supplementary Fig. 7 and Supplementary Data 5). The T cell clusters included a T$_{reg}$ cluster (expressing *FOXP3/IL2RA*), a naïve T cell cluster (*CCR7*) and 4 CTL clusters. A CD8$^+$ exhausted T cell cluster (T-CD8-exhausted) annotated based on expression of exhaustion markers (e.g., *PDCD1* and *LAG3*) was enriched in IE1 (Fig. 2e), as were two CD4$^{high}$ T cell clusters that highly expressed *PDCD1* along with known markers of T follicular helper (Tfh) cells (clusters Tfh-1 and Tfh-2)[39]. Also IE1-enriched were an NKT cell cluster that co-expressed *CD3E* with

classical NK cell markers such as *NCAM1*, *NCR1*, and *KLRC1* (Fig. 2e, f and Supplementary Fig. 7F) and a proliferating cell cluster (T-pro-liferating) that highly expressed *MKI67* along other cell division genes. A single T cytotoxic cluster (3) was significantly more frequent in IE2 environments (Fig. 2e) and overexpressed early activation (*CD69*) and cytotoxicity-mediating transcripts (*TNF*, *GZMK*, *GZMA*) in addition to transcripts encoding a range of ribosomal proteins and TCR components, hinting at weak or transient TCR engagement[40,41]. Enrichment of individual clusters in IE1 or IE2 was not driven by differences in tumor grade (Supplementary Fig. 6B). The proportion of PDCD1-expressing T cells from the scRNA-seq data corresponded well to the proportion of PD-1-expressing T cells from patient-matched CyTOF data (Supplementary Fig. 7G). Our scRNA-seq analysis was also largely confirmed by the IMC data, where we also identified T$_{regs}$, NK cells, CD8$^+$/PD-1$^{low}$, CD8$^+$/PD-1$^{high}$, CD4$^+$/PD-1$^{low}$, and CD4$^+$/PD-1$^{high}$ metaclusters (Supplementary Fig. 8A–C), with CD8$^+$/PD-1$^{high}$ (likely representing exhausted CD8+ T cells) enriched in IE1 tumors. CD4$^+$/PD-1$^{high}$ T cells (likely representing Tfh cells), were not significantly enriched when considering the IMC data alone (Supplementary Fig. 8C).

Next, we explored exhaustion-associated phenotypic T cell states in more detail. T cell exhaustion has been described as a continuous process during which self-renewing progenitor cells express the transcription factor TCF7, which is then gradually lost with increasing exhaustion and decreasing proliferation potential[42]. We indeed observed reduced expression of *TCF7* mRNA in IE1 (Fig. 2b) and specifically in the IE1-enriched T-CD8-exhausted cluster (Fig. 2e, g), although this could not be observed at the protein level in IMC (Supplementary Fig. 8D). IE1 environments were also enriched in proliferating T cells compared to IE2 (Fig. 2e), consistent with IMC analysis (Supplementary Fig. 8E). Additionally, compared to other CD8$^+$ T cell clusters, the IE1-enriched T-CD8-exhausted cluster expressed higher levels of many transcripts that distinguish tumor-reactive T cells from bystander T cells, which do not experience TCR signaling from the tumor[42,43] (Fig. 2g). These include transcripts encoding CD39 (*ENTPD1*), CD103 (*ITGAE*), inhibitory receptors like PD-1 (*PDCD1*), LAG-3, TIM-3 (*HAVCR2*), and CTLA-4, T cell activation markers such as 4-1BB (*TNFRSF9*) and GITR (*TNFRSF18*), and the B cell chemoattractant CXCL13. The T-proliferating cluster was also enriched in all tumor reactivity- and exhaustion-related transcripts and was low in *TCF7* in comparison to non-proliferating cells (all other clusters) (Fig. 2h). Taken together, our results indicate that CTLs expressing exhaustion signatures also express tumor reactive signatures and proliferation markers.

We investigated potential sources of the difference in CD8$^+$ T cell activation in IE1 and IE2. Pseudobulk analysis revealed significantly higher expression of MHC-I-encoding genes in epithelial cells of IE1-classified tumors compared to IE2-classified tumors (Supplementary Fig. 7H), while levels of *CD274* mRNA, which encodes the PD-1 ligand PD-L1, did not significantly differ (Supplementary Fig. 7H, I). This was even more apparent in the IMC analysis, where we observed significantly higher expression of MHC class I proteins (HLA-A, B and C) on IE1 tumor cells (Fig. 2i); this effect could not be explained by tumor grade (Supplementary Fig. 6C). Our data show that MHC class I expression constitutes a major difference between IE1 and IE2 tumor cells and show that, at least in this cohort, MHC-I is better associated with CD8$^+$ T cell activation than PD-L1. Whole-slide immuno-fluorescence analysis of CD8$^+$ T cell infiltration further showed that half

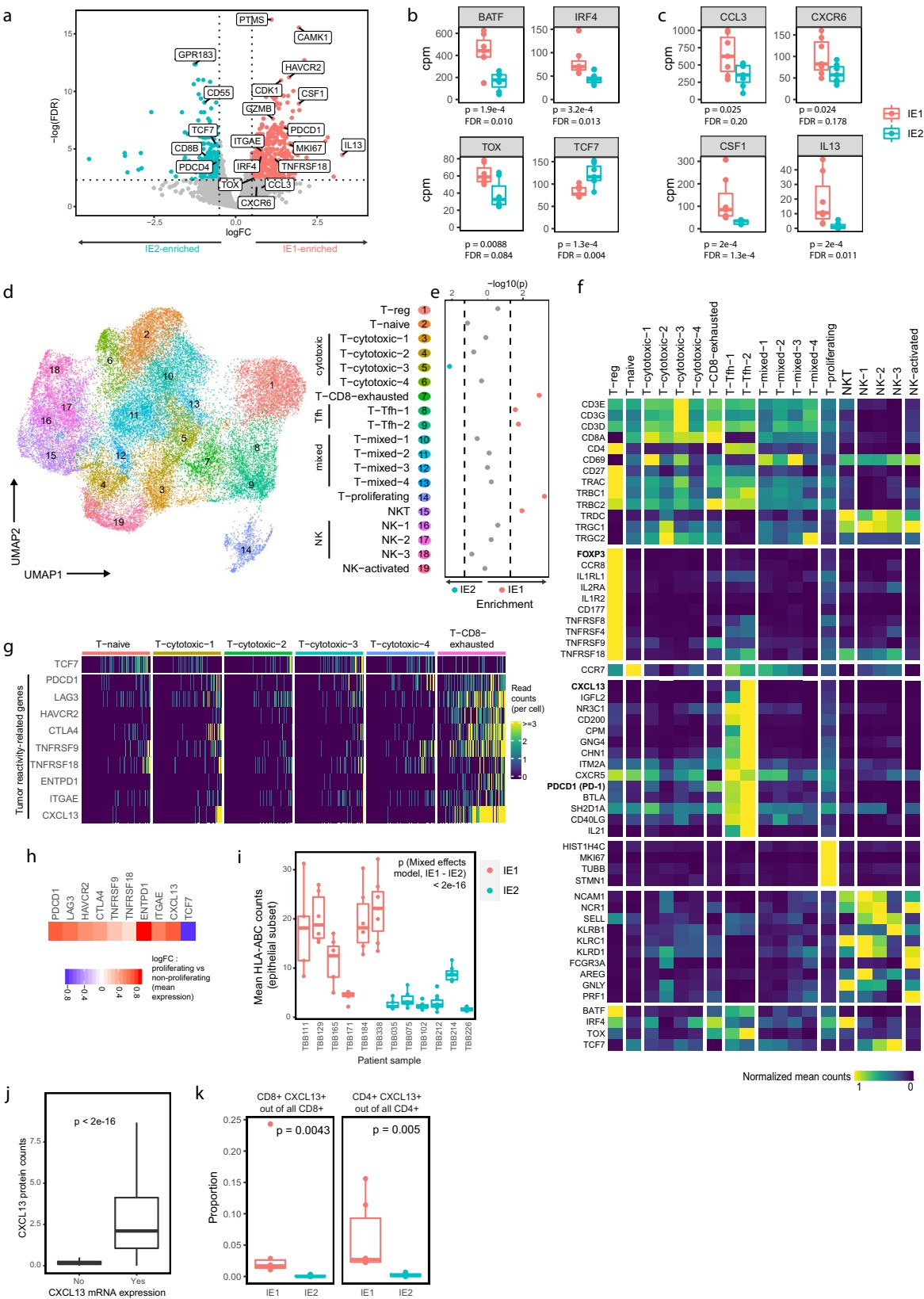

of IE1 tumors, but none of the IE2 tumors, were classified as immune-infiltrated (Supplementary Fig. 7J). This suggests an association of a "hot" immune phenotype with T cell exhaustion, consistent with previous reports[44,45].

Enrichment of antigen-experienced T cells in IE1 was not restricted to the CD8[+] subset: The frequency of Tfh cells, an antigen-experienced

PD-1[high] CD4[+] T cell type known to be involved in B cell maturation and differentiation[46], was higher in IE1 (Fig. 2e) and strongly correlated with the frequency of CD8[+] exhausted T cells (Supplementary Fig. 7K). Tfh frequency was also correlated with plasma cell but not B cell frequency (Supplementary Fig. 7K), indicating that Tfh cells might act as mediators of B cell differentiation into plasma cells in this context. In line

**Fig. 2 | The T cell phenotypic landscape of exhausted and non-exhausted immune environments. a** Volcano plot showing differential expression between T and NK cells of IE1 versus IE2 samples in pseudobulk patient-averaged scRNA-seq data. Dashed lines indicate false discovery rate (FDR) of 0.1 and log2 fold change (logFC) of 0.5. Boxplots comparing T and NK pseudobulk expression in counts per million (cpm) for selected transcription factors (**b**) and cytokines/receptors (**c**) between IE1 and IE2 samples. **d** UMAP plot of scRNA-seq data from 36,000 T and NK cells colored by Seurat cluster, annotated with the indicated cell type labels. **e** Enrichment of cluster frequencies, annotated by cell type, in IE1 or IE2 samples. Dashed lines indicate $p = 0.05$ (two-sided Wilcoxon rank sum test). **f** Heatmap showing normalized average expression of selected marker genes for all T and NK cell clusters. **g** Single-cell count heatmap of selected genes associated with tumor-reactivity and/or exhaustion. 100 cells were randomly sampled from the naïve T cell cluster and from each CD8[+] T cell cluster; columns represent single cells.

**h** Heatmap displaying the fold change in mean expression of the indicated genes in proliferating versus non-proliferating T and NK cells. **i** Boxplot comparing image-averaged single-cell HLA-ABC expression in IMC data for the epithelial subsets of IE1 versus IE2 samples. Each dot represents one image. A mixed effects model was fitted on the log1p-transformed data. **j** Boxplot comparing mean CXCL13 protein counts between *CXCL13*-expressing and non-expressing T cells in IMC. **k** Boxplot comparing CXCL13[high] cell proportions out of all CD8[+] T cells (left) and CD4[+] T cells (right) between IE1 and IE2 samples in IMC. Only non-TLS images were included. For scatterplots, Spearman correlation coefficient (two-tailed test) and $p$ value are indicated. For boxplots, two-sided Wilcoxon rank sum test was used for statistical analysis unless otherwise noted. Boxplot centers indicate group median, bodies show IQR, and whiskers extend to the largest and smallest value lying within 1.5 times the IQR above the 75th percentile and below the 25th percentile, respectively. For **b**, **c**, **i** and **k**: $n = 14$ independent patient samples.

with previous reports, both the T-CD8-exhausted and Tfh cell clusters in our dataset expressed *CXCL13* (Fig. 2e–g), and expression of *CXCL13* and *PDCD1* correlated across clusters and across patients (Supplementary Figs. 7L and 8G). Elevated CXCL13 expression in IE1 was confirmed in IMC on both transcript and protein levels (Fig. 2j, k) and the proportions of CXCL13[+] T cells measured by scRNA-seq and IMC on two separate pieces of the same tumor were highly correlated (Supplementary Fig. 8F), indicating that CXCL13[+] T cell frequency was a tumor-wide characteristic. Our data suggest that CXCL13[+] T cell frequency more clearly distinguished the IE1 and IE2 environments than frequency of PD-1[high] T cells (Fig. 2k and compare Supplementary Fig. 8C) and this difference could not be explained by tumor grade (Supplementary Fig. 6D). Taken together, these findings indicate that T cells in tumors with characteristics of an exhausted immune environment are proliferative, show a tumor-reactive signature and, despite signs of terminal exhaustion, might retain more anti-tumor activity than T cells in tumors with a non-exhausted immune environment.

## Cytotoxic potential in exhausted immune environments is altered but not abolished

Next, we examined how the two immune environments differ in T and NK cell cytotoxic potential, focusing on the main molecules involved in CTL-mediated target cell killing. Pseudobulk comparison of nine selected genes in IE1 versus IE2 samples revealed that the only cytotoxic molecule significantly elevated in the IE2 T and NK cell compartments was *TNF*, and that *GZMB*, *GNLY* (encoding Granulysin), and *FASL* (encoding Fas ligand) were expressed at higher levels in IE1 (Fig. 3a). *GZMB* was amongst the top differentially expressed genes (Fig. 2a), and we also observed this with IMC (Supplementary Fig. 9A). *GZMB* and *FASL* expression were highest in the T-CD8-exhausted cluster (Fig. 3b), indicating that these T cells have the potential to exert cytolytic effector functions despite their exhausted phenotype.

To further assess the difference in cytotoxic potential between IE1 and IE2, we placed the single-cells of the CD8[+] T cell subsets on a linear pseudotime trajectory. We used a Bayesian approach called Ouija[47] to perform pseudotime ordering based on levels of transcripts associated with T cell activation or dysfunction (Supplementary Fig. 9B and Supplementary Data 6). The output was a trajectory with naïve T cells on one end and exhausted T cells on the other, with the cytotoxic clusters intermediate (Fig. 3c, d). The clusters previously identified as transition/early activation phenotypes (T-cytotoxic-3 and -4, Fig. 2) had lower pseudotime scores than the other cytotoxic clusters, supporting the validity of our ordering (Fig. 3d). Not surprisingly, mean pseudotime scores were higher for IE1-classified samples (Fig. 3e), but there was substantial variability within the individual IE groups, supporting the notion of immune exhaustion as a continuum and suggesting that the T cells in the IE1 samples are further along this continuum.

We examined gene expression profiles over pseudotime to identify early- and late-exhaustion- genes, associated with IE2 and IE1 respectively. We discovered that *CXCL13*, *CTLA-4*, and *PD-1* had the

sharpest increases at the late exhausted stage (Supplementary Fig. 9B). *LAG-3* and *TIGIT* displayed a more gradual increase, which also began relatively early in the pseudotime course. Among the cytotoxic genes, we observed an early increase of *IFNG* and *TNF*, closely followed by *GZMB*; but these three markers subsequently followed different trajectories (Fig. 3f). *GNLY* expression was significantly increased in IE1 versus IE2 in pseudobulk analysis (Fig. 3a) and had a pseudotime profile with a peak in the middle and a decrease in the late stages. The main source of *GNLY* expression within the T and NK cell compartments were NKT cells (Fig. 3b), a strongly cytotoxic cell type with a poorly understood role in human cancer[48].

To better understand how NKT cells might act on the IE1 environment, we examined their expression profile in more detail. One of the few uniquely overexpressed genes in the NKT cell cluster was *CSF1* (Fig. 3g), which encodes a cytokine important for the activation and differentiation of myeloid cells[49]. We used this marker to annotate NKT cells in our IMC dataset, defining CSF-1[+] T and NK cells as NKT cells and confirming the overrepresentation of this cell type in IE1 environments at the protein level (Supplementary Fig. 9C).

Thus, the main effectors of T and NK cell-mediated cytotoxicity are distinct in the two immune environments. Our data indicate that the T cells in IE1 and IE2 samples are present in an exhaustion continuum, and that IE1 T cells are further along this continuum than those in IE2. IE1 is characterized by strong upregulation of *GZMB* in exhausted T cells and the presence of *GNLY*-expressing NKT cells, suggesting cytotoxic potential. The previously unreported expression of CSF-1 by NKT cells in immune environments enriched in exhausted T cells may represent a link between T cell activity and myeloid-cell-mediated immune responses.

## Myeloid cells in exhausted immune environments indicate increased inflammation

We next analyzed myeloid cell subsets in IE1 and IE2. Pseudobulk analysis on the myeloid cell fraction (monocytes, macrophages and DCs, excluding granulocytes) (Fig. 4a) showed that several cytokine-encoding mRNAs were overexpressed in IE1 (Supplementary Fig. 10A), most notably *CCL18*, which has previously been associated with breast cancer metastasis[50]. Genes encoding major complement system components were more highly expressed in IE1 compared to IE2 (Supplementary Fig. 10A), indicating higher inflammatory and phagocytic potential of IE1 myeloid cells. In IE2-classified tumors, we observed higher expression of *CD55* and *CD46*, which encode proteins that limit complement function[51], in T and NK cells (Supplementary Fig. 10B). Finally, IE1 myeloid cells showed increased expression of transcripts encoding matrix metalloproteinases, implicated in breast cancer invasion and metastasis[52], and of metallothioneins, which protect cells from oxidative stress and cytotoxicity (Supplementary Fig. 10A). Running the same analysis while accounting for tumor grade revealed a similar trend (Supplementary Fig. 6E). Taken together, these data are indicative of inflammation in IE1 tumors.

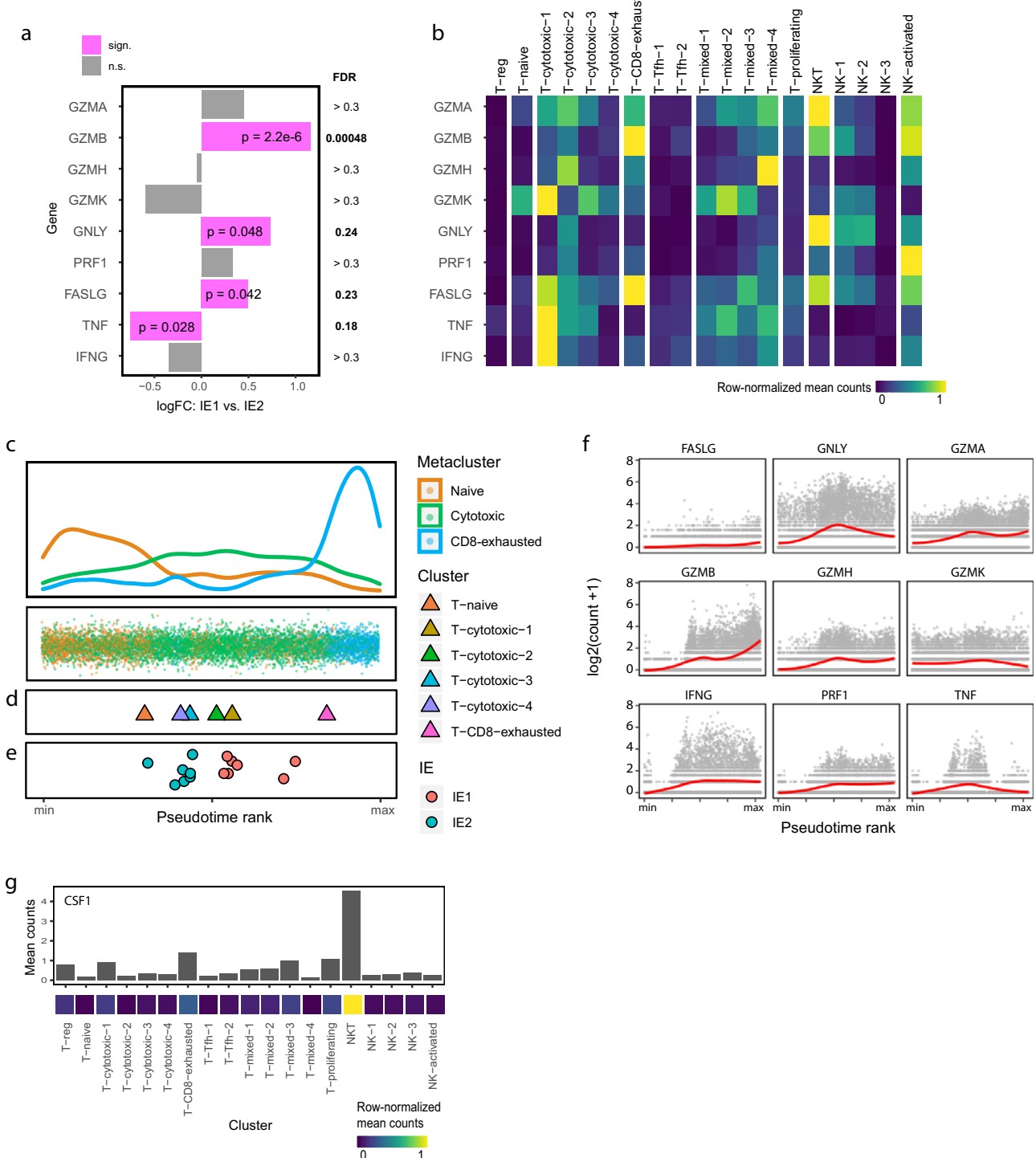

**Fig. 3 | Cytotoxic effector profiles differ between exhausted and non-exhausted immune environments. a** Bar plot showing differential expression of the indicated transcripts in T and NK cells between IE1 and IE2 tumors (patient-averaged pseudobulk data). *p* values are derived from EdgeR analysis and are not multiple testing corrected. FDR values indicate the genome-wide false discovery rate as given by EdgeR. **b** Heatmap showing normalized average single-cell expression of cytotoxic genes for all T and NK cell clusters. **c** Pseudotime ordering of CD8+ T cells in all samples based on scRNA-seq data. Single cells are colored according to metacluster (bottom) and the corresponding density plot is displayed (top). **d** Mean pseudotime scores for individual cell phenotype clusters. **e** Mean pseudotime scores for individual samples colored by immune environment. **f** Single-cell expression of the indicated cytotoxic genes along pseudotime. The analysis was done on scRNA-seq data from CD8+ T cells in all samples. Red line corresponds to locally estimated scatterplot smoothing (LOESS) curve. **g** Average single-cell *CSF1* expression in all T and NK cell clusters displayed as a bar chart (top) and in a normalized heatmap (bottom).

Graph-based subclustering of the scRNA-seq profiles from the myeloid cell fraction revealed two clusters of monocytes or early differentiating macrophages (mono-1 and mono-2), seven clusters of TAMs, five clusters of DCs, and one cluster of proliferating myeloid cells (Fig. 4b and Supplementary Fig. 10C). All patient samples contained monocytes, TAMs, and DCs (Supplementary Fig. 10D), although three TAM clusters were found mainly in one sample each (Supplementary Fig. 10E). Most monocyte and TAM clusters tended to be

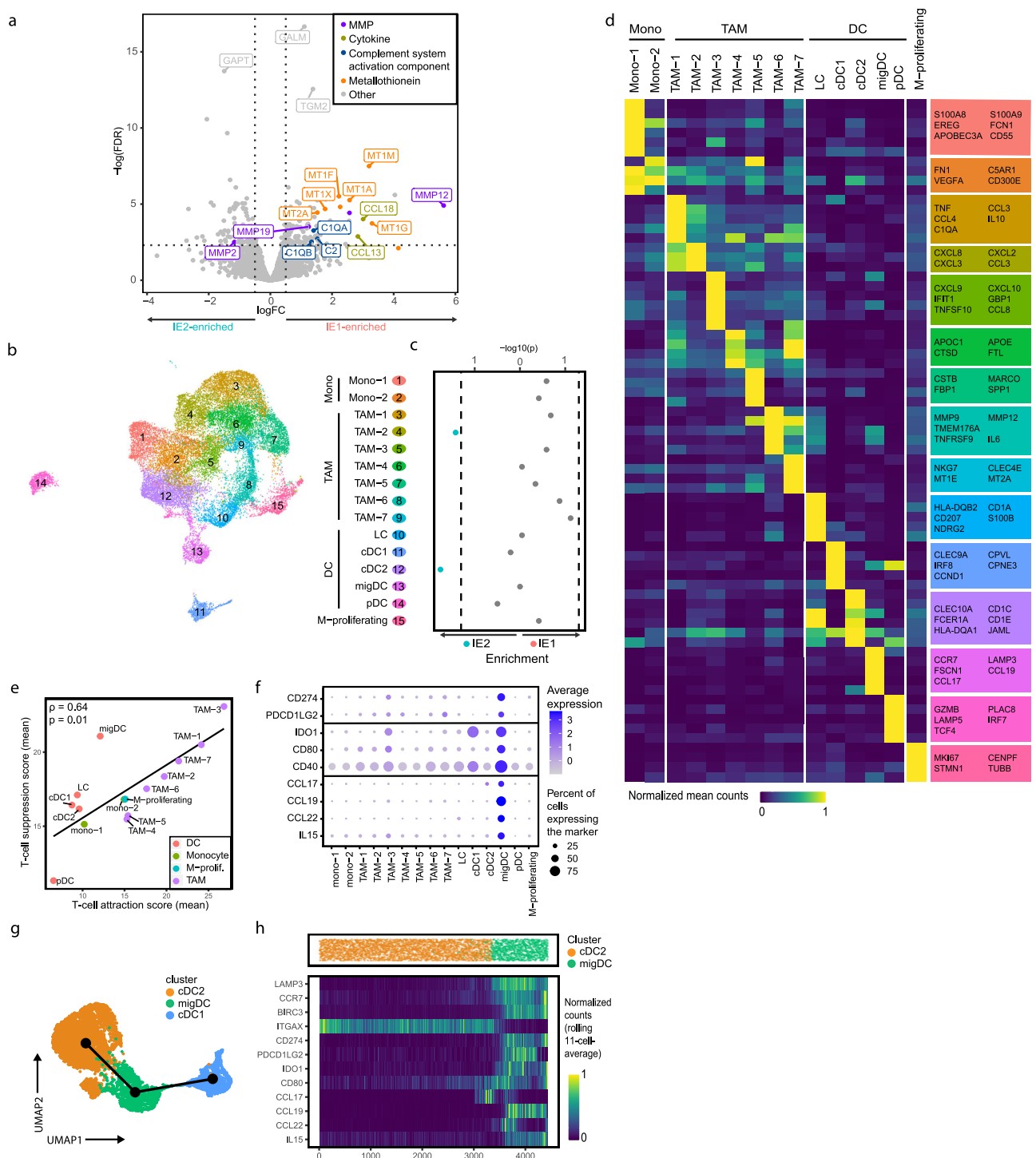

**Fig. 4 | Myeloid cell phenotypes in exhausted immune environments indicate inflammation and T cell-suppressive potential. a** Volcano plot showing differential gene expression between myeloid cells of IE1 and IE2 samples in pseudobulk patient-averaged scRNA-seq data. Dashed lines indicate an FDR of 0.1 and a logFC of 0.5. Genes are colored by functional group. **b** UMAP plot of scRNA-seq data from 26,000 myeloid cells colored by Seurat cluster and annotated by cell type. **c** Enrichment of cluster frequencies in IE1 and IE2 samples. Two-sided Wilcoxon rank sum test was used for statistical analysis and dashed lines indicate a *p* value of 0.05. **d** Heatmap showing normalized average single-cell expression of the top 10 differentially expressed genes for all myeloid cell clusters. Selected genes

overexpressed in the respective cluster are indicated in the colored boxes. **e** Scatterplot of the mean T cell-suppression score versus the mean T cell-attraction score for all myeloid cell clusters. **f** DotPlot showing expression of main migDC markers across all myeloid cell clusters. **g** UMAP of cDC subsets and migDCs with Slingshot trajectories overlaid. **h** Slingshot pseudotime ordering of single cells from the cDC2 and migDC subsets (top) and heatmap showing normalized expression of selected genes along pseudotime using the rolling average expression over 11 cells (bottom). Genes with log counts per million <1.5 in EdgeR analysis were excluded for plots **a**–**d**. For scatterplots, Spearman correlation coefficient (two-tailed test) and *p* value are indicated.

more frequent in IE1 tumors without any individual cluster reaching significance (Fig. 4c), whereas IE2 tumors showed enrichment of cluster TAM-2 as well as classical DCs type 2 (cDC2). Enrichment of cDC2 in IE2 was not driven by differences in tumor grade (Supplementary Fig. 6F). Cells of the mono-1 cluster showed a monocyte-specific gene signature, while mono-2 cells showed patterns overlapping with several TAM clusters and likely represent a transition phenotype from monocyte to mature macrophage (Fig. 4d). TAM clusters showed high expression of *CD68*, *CD163*, *MRC1 (*also known as *CD206*), and *MSR1* (also known as *CD204*) (Supplementary Fig. 10F).

We next examined crosstalk between T cell and myeloid compartments. We quantified T cell-attractive and T cell-suppressive properties of myeloid cells using gene signatures that we assembled from the literature (Supplementary Data 6); spatial IMC analysis further supported the T cell attraction signature and a subset of markers within the T cell suppression signature (Supplementary Fig. 10G, H). The two signatures showed a positive correlation at the single-cell level (Supplementary Fig. 10I), at the patient level (Supplementary Fig. 10J), and at the level of phenotypic clusters (Fig. 4e), suggesting that myeloid cells that harbor a strong suppressive potential might also actively attract T cells. Most TAM clusters showed high scores for both signatures, whereas monocyte and DC clusters tended to have low scores. Both signatures tended to be enriched in IE1 tumors compared to IE2 tumors, as were gene signatures for "classically activated" (M1) and "alternatively activated" M2 macrophages[27] (Supplementary Fig. 10K, L), further supporting an overall higher activation and differentiation state of myeloid cells in IE1 versus IE2 tumors.

We also observed markers of T cell suppression among DCs. Apart from cDC2s, we distinguished four other DC subsets previously described in human breast tumors[52] (Fig. 4d). Classical DCs type 1 (cluster cDC1), Langerhans cells (cluster LC) and plasmacytoid DCs (cluster pDC) were identified based on their high expression of *CLEC9A*, *CD207*[53] and *IRF7/PLAC8/IL3RA*, respectively. We further identified a DC subset with elevated expression of *LAMP3*, *FSCN1*, and *CCR7* as migratory DCs (migDCs), as described recently in healthy thymus and various cancer types including breast[15,54–56]. The migDC cluster displayed a particularly strong T cell-suppression signature compared to other DCs (Fig. 4e). migDCs showed the highest expression of *CD274* (which encodes PD-L1) in the entire TIME (Fig. 4f and Supplementary Fig. 11A), and were also the main producer of *PDCD1LG2*, which encodes the second known PD-1 ligand PD-L2. The migDCs also expressed *IDO1*, *CD80*, and *CD40* and exhibited a unique cytokine expression profile with high counts of *CCL17*, *CCL19*, *CCL22*, and *IL15*, which were not detected in other myeloid cell subsets. IMC confirmed co-expression of PD-L1, IDO1, and CD40 proteins and *CCL17* and *CCL22* transcripts with the migDC marker *LAMP3* (Supplementary Fig. 11B, C). The expression profile of migDCs suggests the potential to recruit immune cells and contribute to T cell suppression; however, the abundance of migDCs did not differ between IE1 and IE2. Slingshot trajectory inference analysis implied a transition from cDC2s to migDCs, consistent with previous work[56,57]; the cDC1 subset clustered apart (Fig. 4g). Using Monocle2 for trajectory inference gave similar results (Supplementary Fig. 11D). We observed distinct expression dynamics of the migDC markers such as *CCL17*, *CCL22*, and *CCL19* along the cDC2-migDC transition (Fig. 4h).

In summary, immune environments with and without enrichment of exhausted T cells harbored myeloid cells that differed in their inflammatory states, and we observed evidence of crosstalk between myeloid and T cells. We have identified a LAMP3+ migratory DC subset that likely originated from cDC2 cells and expressed markers indicating high T cell-suppressive potential.

## Immune states are linked to distinct cell-to-cell communication patterns

We systematically investigated how cellular interaction differs in the IE1 and IE2 immune environments using SingleCellSignalR, an algorithm that infers intercellular networks from single-cell transcriptomic data based on a manually curated ligand-receptor (LR) database[58]. We first quantified the total number of predicted LR interactions for each cell type pair across all samples (Fig. 5a and Supplementary Fig. 12A and Supplementary Data 7). This revealed many predicted interactions between fibroblasts and endothelial cells, and also myeloid cell auto-interactions. T cells, B cells, and plasma cells were generally predicted to have a lower number of interactions, partly explicable by the overall fewer expressed genes in these cell types. Analysis with a second algorithm for inference of cellular crosstalk, CellPhoneDB, confirmed these trends (Supplementary Fig. 12B and Supplementary Data 8). To separate cell type-specific from ubiquitous interactions, we used the coefficient of variation of the 100 top-scoring predicted LR interactions for each of the main cell type pairs and thus identified pair-specific interactions (Supplementary Fig. 12C). We found that, for example, fibroblast-to-endothelial interactions were dominated by collagens and integrin receptors, whereas specific myeloid-to-T cell crosstalk involved interactions of several cytokines with their cognate receptors.

We next compared predicted interactions in breast tumor immune environments with and without evidence of T cell exhaustion. We applied SingleCellSignalR to data from each patient individually (focusing on epithelial, myeloid, and T and NK cell interactions) and found a number of predicted LR pairs enriched in either IE1 or IE2 tumors (Fig. 5b). As expected, many predicted immunomodulatory interactions such as *CD274* with *PDCD1* (at the protein level, PD-L1 with PD-1), *CD80* with *CTLA-4*, *LGALS9* with *HAVCR2* (at the protein level, Galectin-9 with TIM-3), and *PVR* with *TIGIT* were enriched in IE1 tumors, with mRNAs encoding the ligands for PD-1 and CTLA-4 expressed at significantly higher levels only by myeloid cells. IE1 tumors also had increased scores for chemotactic interactions, mainly mediated by CCL3, CCL4, CCL5, and CXCL9, suggesting a more dynamic immune environment with increased potential for recruitment of cells from outside the tumor. Furthermore, several predicted non-chemotactic cytokine interactions were enriched in IE1 tumors, some activating (CSF-1 and IL15 with respective receptors), and others with more complex predicted functions (e.g., IL10 with its receptor). The only predicted cytokine interaction enriched in IE2 tumors was that of FLT3 ligand, a stimulator of DC growth[59] and expressed by T and NK cells and its receptor on myeloid cells. Other interactions enriched in IE2 tumors included those of ERBB4 and of thrombospondin with their ligands. A comparable analysis with CellPhoneDB yielded similar results (Supplementary Fig. 12D).

To examine how specific myeloid, T and NK cell subtypes may interact with each other, we repeated the interaction analysis focusing only on these subtypes (Fig. 5c). For better biological interpretability, we first aggregated the identified clusters into larger metaclusters, each representing a specific cellular subtype. All T and NK cell and myeloid metacluster pairs had between 300 and 600 predicted interactions, with TAM auto-interactions the most frequent. With the exception of migDCs, cell types with many outgoing interactions (i.e., the cells express the ligand) also had many incoming interactions (i.e., the cells express the receptor). Predicted LR interactions that were specific for certain metacluster pairs (Supplementary Fig. 12E) included the interaction between Tfh cell-specific IL21 and IL13 with respective receptors on myeloid clusters, the interaction between CCL18 and CCR8 (on TAMs and $T_{regs}$, respectively) and that of CCL17 and CCL22 (expressed by migDCs and cDC2s) with CCR4 (expressed predominantly by $T_{regs}$ and Tfh).

We then predicted which myeloid-derived ligands are the most closely linked to T cell exhaustion using NicheNet, a method that builds upon prior knowledge to predict which ligands of the sender cells (in our case, myeloid cells) are most likely to have affected the expression of known target genes (in our case, CD8+ T cell exhaustion-related genes) in the receiver cells. The myeloid-derived ligand that

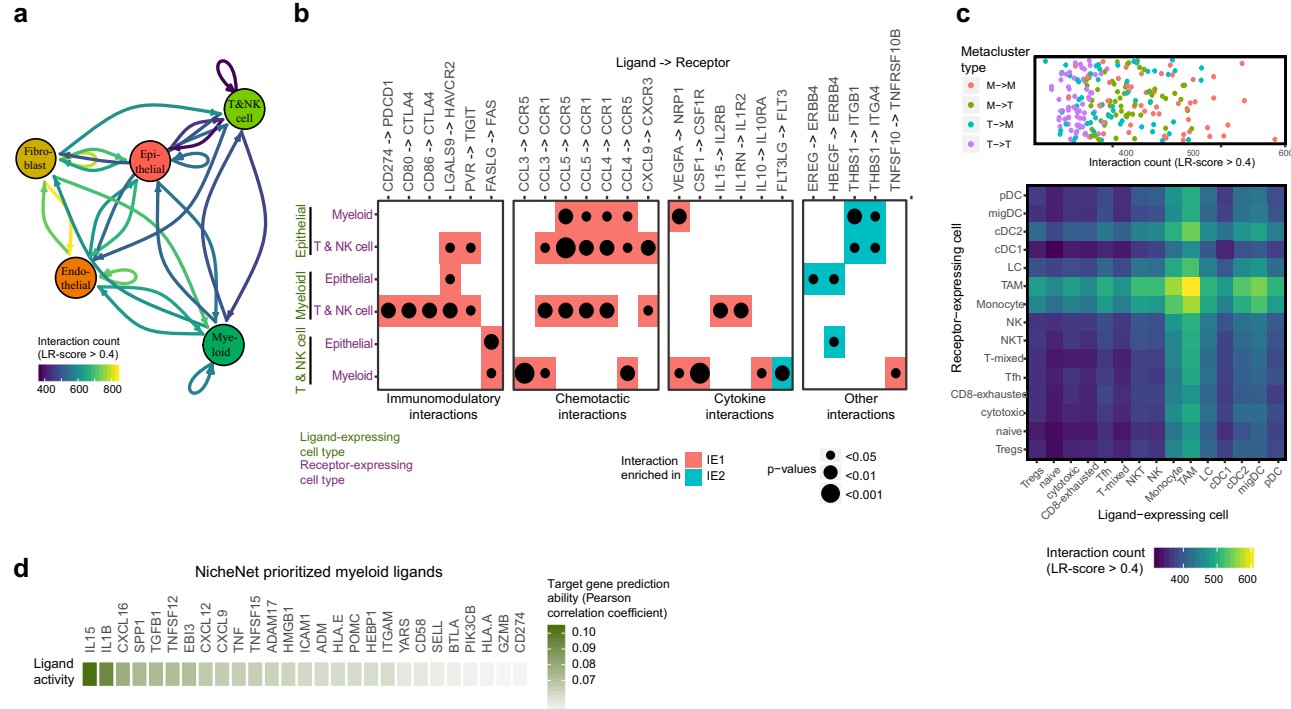

**Fig. 5 | Ligand-receptor analysis predicts TIME-wide and exhaustion-specific cellular crosstalk. a** Social graph depicting the number of interactions between the five most frequent cell types. **b** Enrichment of selected ligand-receptor interactions in either IE1 or IE2 tumors for the given cell type pairs. Selections were made based on literature evidence and biological interpretability. The full list of enriched interaction pairs is in Supplementary Data 6. White squares denote interactions with an enrichment *p* value >0.05 or a mean LR score <0.4 in the given cell type pair.

Two-sided Wilcoxon rank sum test was used for statistical analysis. **c** DotPlot (top) and heatmap (bottom) depicting the number of interactions between different myeloid and T and NK cell metaclusters. M indicates myeloid metacluster; T indicates T and NK cell metacluster. **d** Heatmap showing the myeloid-derived ligands with the highest ability to affect exhaustion-related target gene expression as predicted by NicheNet.

best predicted expression of exhaustion-related genes was IL15, followed by IL1B and CXCL16 (Fig. 5d), consistent with enrichment of IL15-IL15 receptor interaction in IE1 versus IE2 tumors (Fig. 5b and Supplementary Fig. 12D). The main producers of IL15 were migDCs (Fig. 4f), providing yet another link between migDCs and T cell exhaustion. We note that we cannot however rule out that myeloid cells are not only upstream but also downstream of exhausted T cells.

In conclusion, LR analysis provided a comprehensive quantitative map of potential cellular interactions in breast cancer TIMEs. We report an enrichment of predicted T cell-regulatory interactions and evidence for enhanced cytokine and chemokine signaling, especially from myeloid cells to T and NK cells, in immune environments high in exhausted-like T cells.

**Spatial distribution of immune cells varies with exhaustion**

LR analysis can predict intercellular communication but it is blind to spatial proximity, a key requirement for cellular interaction. We therefore used our IMC data (Supplementary Data 9) to ask whether cells that we predicted to interact are also physically proximal in the tissue and whether spatial patterns differ in IE1 and IE2.

We used pairwise neighborhood analysis to quantify relative avoidance or interaction for each cell type pair (accounting for cell type frequency)[60] and found that fibroblasts and endothelial cells were strongly enriched in each other's neighborhood (Fig. 6a) (box 1), consistent with predicted LR interactions between the two (Fig. 5a). Similarly, spatial analysis is consistent with predicted myeloid cell auto-interactions (box 2). As expected, tumor cell subtypes formed a spatial cluster (box 3) and tended to avoid other cell types. However, some tumor subtypes did show spatial proximity to other cell types: hypoxic and apoptotic tumor cells had a higher likelihood of neighboring immune cells (box 4) and PD-1$^{high}$ T cells were more likely to

border tumor cells than PD-1$^{low}$ T cells (box 5). We also observed spatial patterns within immune cell types. Myeloid cells, T$_{regs}$, CD4$^+$ T cells, and CD8$^+$ T cells formed a spatial cluster (box 6), as did PD-1$^{high}$/CD8$^+$ T cells, PD-1$^{high}$/CD4$^+$ T cells, and migDCs (box 7).

Overall, PD-1$^{high}$ T cells were significantly more likely to have at least one migDC as their direct neighbor compared to PD-1$^{low}$ T cells (Fig. 6b, c). This is consistent with our predicted PD-1-PD-L1 interaction between myeloid and T cells and with the finding that migDCs were the main producer of PD-L1 in our samples, and suggests direct engagement of PD-L1-expressing migDCs with PD-1-expressing T cells. To further examine exhaustion-associated spatial motifs, we quantified cell type composition of the immediate neighborhood (10 closest neighbors) of different T cell subtypes in images that did not contain TLS. This revealed that, on average, PD-1$^{high}$ T cells were surrounded by fewer myeloid cells and fibroblasts, but by more other PD-1$^{high}$ T cells, pDCs and migDCs, when compared to PD-1$^{low}$ T cells and T$_{regs}$. The same trends were seen for images with TLS (both mature and immature) (Fig. 6d).

Pairwise neighborhood analysis did not reveal differences between the relative interaction or avoidance of cell type pairs in IE1 compared to IE2. However, IMC showed a more than five-fold higher proportion of cytokine-expressing cells in IE1 than in IE2 tumors especially for T and NK, SMA$^+$ stromal cells, and myeloid cells (Supplementary Fig. 13). In addition, a large proportion of cytokine-expressing cells in IE1 but not IE2 tumors were part of a cytokine patch (defined as a spatial cluster of at least three cytokine-expressing cells directly neighboring each other) (Supplementary Fig. 13C). To identify the cell types surrounding these patches across the whole cohort, we defined cytokine milieus comprising all cells within a radius of 30 μm from a given cytokine patch and quantified cell types in these milieus compared to the overall tissue (Fig. 6e). CD4$^+$ T cells were enriched in

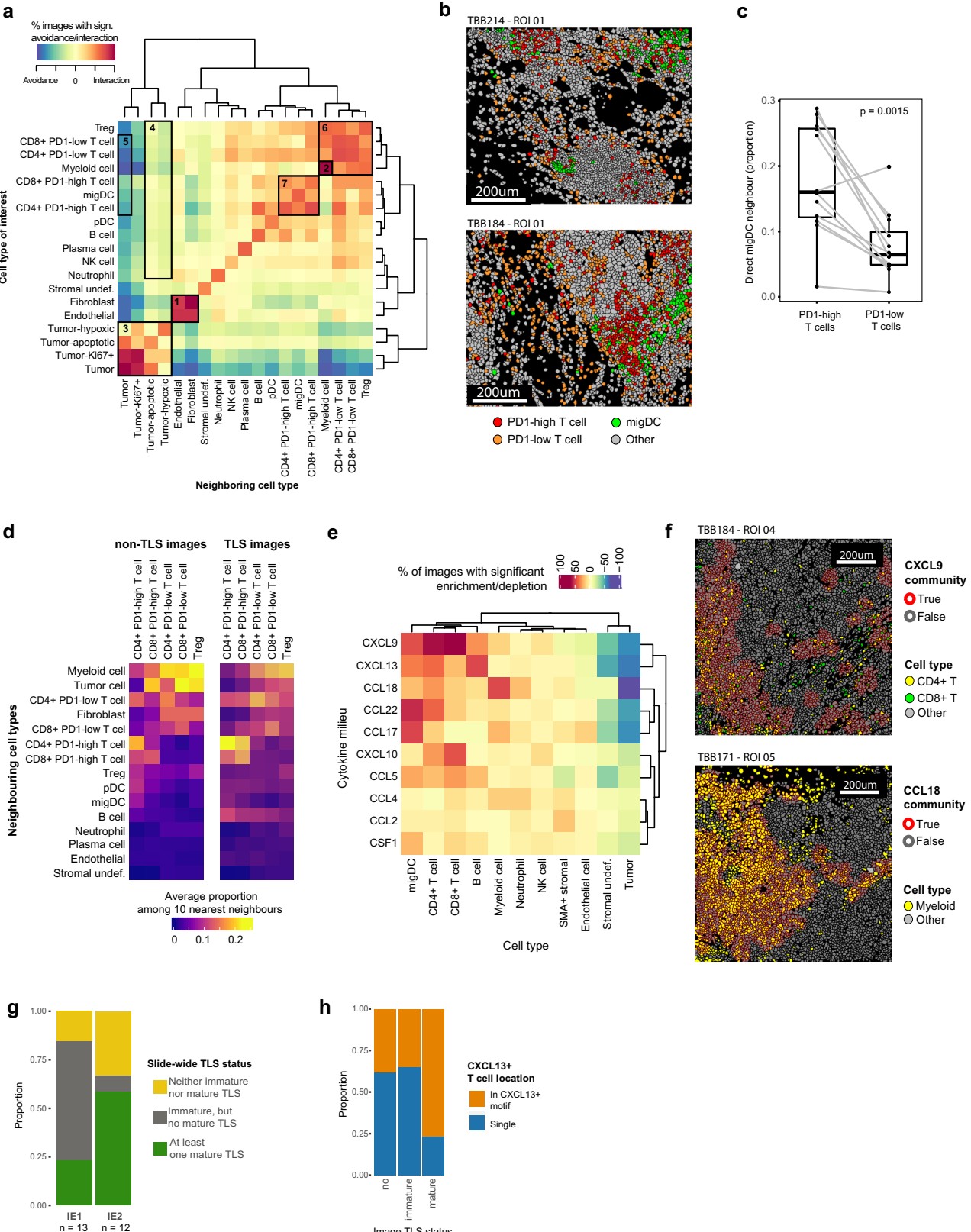

*CXCL9, CCL22* and *CXCL13* milieus, whereas CD8+ T cell enrichment was highest in *CXCL9* and *CXCL10* milieus (Fig. 6f, top). As expected, B cells were strongly enriched in *CXCL13* milieus, migDCs in *CCL17* and *CCL22* milieus, and myeloid cells in *CCL18* milieus (Fig. 6f, bottom). Tumor cells were depleted across most cytokine milieus. We conclude that spatial cytokine expression patterns differ in tumor environments with

and without evidence of T cell exhaustion, and that these expression patterns in turn are linked to immune cell type distribution.

One of the most distinct structural elements in solid tumors are TLS, which differ considerably from the surrounding tissue in cell type frequency and spatial distribution[61]. Our transcriptomics data showed higher frequency of Tfh cells and lower frequency of B cells in IE1

**Fig. 6 | IMC reveals cellular neighborhoods, cytokine milieus and tertiary lymphoid structures. a** Heat map indicating significant pairwise cell type interaction/avoidance in individual images from the Protein Panel dataset ($n = 77$ images, 1000 permutations each). Significance is indicated by square color ($p < 0.01$, two-sided permutation tests), corrected for relative cell type frequency. Highlighted interactions indicate (1) fibroblast-endothelial interactions, (2) myeloid auto-interactions, (3) tumor compartment, (4) hypoxic/apoptotic tumor cell to immune cell interactions, (5) tumor to T cell subtype interactions, (6) main immune compartment, and (7) migDC-PD-1[high] T cell interaction. **b** Single-cell masks for selected IMC images (top: mature TLS image, bottom: immature TLS image). Only a subsection of each image is shown. **c** Paired boxplot comparing the percentage of PD-1[high] T cells versus PD-1[low] T cells that have at least one migDC as a direct neighbor. Each pair of dots represents a separate sample ($n = 14$ independent patient samples; two-sided paired Wilcoxon rank sum test). Boxplot centers indicate group median, bodies show IQR, and whiskers extend to the largest and the smallest value lying within 1.5 times the IQR above the 75th percentile and below the 25th percentile, respectively. **d** Heatmap displaying the average relative proportion of each indicated cell type among the 10 nearest neighbors for each T cell subtype across non-TLS images (left) and TLS-images (mature and immature, right). **e** Heatmap indicating significant relative enrichment or depletion of each cell type in the different cytokine milieus in all images of the RNA Panel dataset (Fisher's exact test; $n = 77$ images; for each individual combination, only images containing the respective community and cell type were included). **f** Single-cell masks for selected representative IMC images with cell outline colored by the indicated cytokine community and cell body colored by cell type. The selected images are representative of cellular patterns seen across the dataset ($n$ total $= 77$ images). **g** Stacked barplots indicating the slide-wide TLS status for 13 IE1 samples and 12 IE2 samples. **h** Stacked barplots showing the proportions of *CXCL13*[+] T cells that are part of a CXCL13-cytokine-cluster for images with the indicated TLS status.

versus IE2 tumors (Supplementary Fig. 7L), and was therefore contradictory for the likelihood of TLS in these two groups. To directly examine TLS in both immune environments, we acquired immunofluorescence whole-slide scans of 13 additional samples from the original cohort[29] and from 12 of the samples used in this study. We found that densely packed, mature TLS were more frequent in IE2-classified tumors, whereas immature TLS (see methods for criteria, and Supplementary Fig. 2B) were more frequent in IE1-classified tumors (Fig. 6g). In IMC, images containing TLS were enriched in most immune cell types compared to images that did not contain TLS. With the exception of B cells, there were only minor differences in immune cell frequencies between images with immature versus mature TLS (Supplementary Fig. 13D). Although this was also true for *CXCL13*[+] T cells, which reportedly play a role in TLS formation[62] (Supplementary Fig. 13E), the spatial distribution of *CXCL13*[+] T cells changed between images containing mature versus immature TLS; specifically, most *CXCL13*[+] T cells were part of a CXCL13-cytokine patch in images containing mature, but not immature TLS (Fig. 6h). Neighborhood analysis on images containing any TLS further revealed an enrichment of direct PD-1[high] T cell-migDC interactions in these images compared with the full dataset (Supplementary Fig. 13F, compare to Fig. 6a).

In conclusion, spatial analysis was consistent with major predicted intercellular interaction axes and revealed that spatial clusters of cytokine-expressing cells were frequent in IE1 but not IE2 tumors. Our data suggest that migDC-mediated regulation of T cell activity occurs via direct interactions that take place preferentially in regions at or near mature and immature TLS and that tumors with different immune environments differ in the spatial distribution of *CXCL13*[+] T cells.

## Discussion

Despite having a better prognosis than triple-negative and HER2[+] tumors, breast cancer of the luminal subtype causes more deaths due to its high incidence[63]. Immune-checkpoint therapy for luminal B tumors is currently being evaluated in clinical trials but has not been approved[64]. To better understand the responses of breast cancer patients to immune modulators, we have performed detailed single-cell transcriptomic and spatial proteomic characterization of distinct breast tumor immune environments at single-cell resolution. These analyses revealed systematic changes in immune environments that harbor large numbers of exhausted-like T cells relative to environments without such cells. We found that the presence of a PD-1[high] exhaustion-like T cell phenotype was associated with an inflamed immune environment with altered cytotoxic potential, cellular composition, intercellular crosstalk, and spatial organization.

In CD8[+] T cells that express markers of exhaustion, hallmark genes of tumor-reactive T cells were upregulated, consistent with results in other tumor types[43]. Strikingly, expression of MHC-I transcripts in tumor cells in IE2-classified tumors, which have few exhausted-like T cells, was substantially lower than in IE1-classified tumors, which

could explain the reduced tumor reactivity of T cells in IE2 tumors[65]. We note that differential MHC-I expression by tumor cells is probably not the only reason for higher T cell stimulation in IE1-classified tumors. Indeed, transcripts for a number of antigens such as CTAG2, MAGEA3, and MAGEA6, which are normally expressed in testis and are immunogenic when aberrantly expressed in cancer tissue, were strongly expressed in a subset of IE1, but not IE2, samples (Supplementary Data 5). Although we do not have quantitative data on tumor mutational burden or neoantigen load for our cohort, these factors have also been linked to T cell activation in breast and other cancer types[62,66,67].

Contrary to the classical notion of T cell exhaustion but in line with findings in other cancer types[13,14], T cells with characteristics of exhaustion expressed markers of proliferation. Differential gene expression analysis revealed that the cytotoxic effector potential of these exhausted-like cells was most probably altered but not abolished. Pseudotime analysis suggested a decrease of *TNF* but an increase of *GZMB* and *FASL* expression during the progression to T cell exhaustion, indicating a shift in the cytotoxic profile from classical inflammatory cytokines to molecules mediating cytolytic effector functions. In contrast to reports from other tumor types[13,14], we did not observe an increase of IFNG expression associated with T cell exhaustion. The cytotoxic potential of IE1 environments was further enhanced by the presence of NKT cells, which highly expressed *GNLY* and *GZMA*; these cells were almost absent from IE2 tumors. However, our data cannot tell whether the relationship of NKT cells with T cell exhaustion is causal. In addition, NKT cells expressed the cytokine *CSF1*, and might also contribute to myeloid cell activation and maturation, thus suggesting a functional link between the lymphoid and the myeloid lineages.

The transcriptional profile of myeloid cells in environments harboring exhausted T cells revealed more inflammatory phenotypes, including a larger number of cytokine-expressing cells, compared to environments without signs of T cell exhaustion. This increased tissue inflammation was accompanied by overexpression of matrix-remodeling metalloproteinases, a phenotype linked to higher risk of metastasis[52,68]. Thus, although T and NK cells in exhausted immune environments probably retain cytolytic capacity, this is not sufficient to stop disease progression and might even contribute to tumor invasiveness via inflammation-induced tissue remodeling.

Our analysis revealed an altered spatial organization of immune environments in breast tumors that did and did not contain exhausted-like T cells. First, we observed that, in tumors with exhausted-like T cells, cytokine-expressing cells were more frequently organized into spatial patches than in tumors without such T cells. Second, we observed an intriguing difference in the frequency of TLS, structures that are associated with favorable outcome in many cancer types although their prognostic value in breast cancer remains unclear[69–73]. Our analyses revealed that antigen-experienced CD8[+] and CD4[+] T cells

highly expressed the chemokine CXCL13, a B cell attractant essential for TLS formation[62]. Surprisingly, elevated CXCL13 expression in IE1 tumors did not translate into higher frequency of TLS. Instead, we observed more sites of loosely accumulating B cells and CXCL13[+] T cells that lacked the dense core typical for TLS structures and might represent either immature or dissolving TLS. Contrary to mature TLS, where CXCL13[+] T cells tended to occur in spatial clusters, CXCL13[+] T cells in these immature sites often were spatially apart from each other. In addition, occurrence of CXCL13[+] T cells outside of immature or mature TLS was a defining characteristic of tumors that harbored exhausted-like T cells. This suggests that the immune microenvironment of these exhausted-like, IE1-classified tumors is more dynamic than that of IE2-classified tumors, with T cell-mediated signals continuing to attract more immune cells. Consistent with this, our data showed a systematic enrichment of immunomodulatory, chemoattractive, and cytokine signaling in exhausted versus non-exhausted immune environments.

Overall, our single-cell comparison of tumors with and without markers of T cell exhaustion has revealed that there is likely to be a more widespread functional and organizational distinction between the corresponding immune environments. Notably, several of our main conclusions hold true upon integration of our scRNA-seq dataset with those from two previous studies of breast cancer[15,57] (Supplementary Fig. 14). In all three datasets, tumor cell MHC-I expression was correlated with the proportion of exhausted T cells, and there was also a correlation between proliferating and exhausted CD8[+] T cells. Classification of samples from the previous datasets into IE1 and IE2 based on the proportion of exhausted T cells showed that, as in our own data, CXCL13, Granzyme B, and to some extent CSF-1 were differentially expressed between these sample types. In general, there was a good correspondence in the immune cell phenotypes in all three datasets (Supplementary Fig. 14).

In our cohort, we identified a subset of myeloid cells as migDCs, an activated cell type that has been recently described in healthy thymus and in a number of cancer types including breast cancer[14,15,54,57]. Our data suggest that the role of these *LAMP3*[+] DCs in the TIME is complex: high levels of cytokine expression suggests recruitment and activation of other immune cells, but migDCs also showed high T-cell-suppressive potential, underlined by high expression of *IDO1* and *CD274* (which encodes the PD-1 ligand PD-L1) and direct spatial interaction with PD-1[high] T cells. Although PD-L1 expression in the TIME is associated with T cell exhaustion, migDC frequency was comparable in IE1- and IE2-classified tumors. Pseudotime analysis indicated that most migDCs in our cohort originate from cDC2, a cell type significantly enriched in IE2 environments, which might counterbalance the expected overrepresentation of migDCs in exhausted (IE1) environments. The migratory phenotype predicted by the migDC transcriptional profile was previously validated in vitro, and showed that migration is lymph-node directed[14]. Thus, we speculate that migDCs might not accumulate at the tumor site but may migrate to the draining lymph node upon full maturation, providing an explanation for the similar migDC frequencies observed in IE1 and IE2 tumors.

Despite rigorous method validation and careful cohort selection, our study has limitations: First, we did not study the third tumor-immune group (TIG) identified in our previous study[29] (representing 31% of all tumors), as this group had generally a small immune proportion and few T cells compared to the other two groups. Second, the low number of cells obtained from most breast tumors made additional experiments such as ex vivo validation of T cell cytotoxicity impossible. Third, the limited sample number constrains the statistical power and might increase the chance to observe spurious relationships between variables in certain cases; this is particularly true in case of relatively rare cell types such as the NKT cells we found enriched in samples with evidence of exhausted-like T phenotypes. Fourth, while

the samples included in our study were mostly luminal, they did include a single triple-negative and a single luminal B-Her2[+] subtype; again due to small sample number, we did not consider subtypes in our analysis. Finally, histological tumor grade is a covariate that is not perfectly balanced in our cohort. However, we found that the characteristics that differed most significantly between IE1 and IE2 showed no difference between low-grade and high-grade tumors in our cohort, and that accounting for grade in pseudobulk analysis did not substantially change the observed differential expression pattern.

PD-1 expression has previously been associated with poor prognosis in luminal B tumors[74], but because data on disease progression and survival is not yet available for our cohort, the impact of IE classification on patient outcome remains to be analyzed. However, it has recently been demonstrated that a high abundance of PD1[high]/CXCL13[+] CD4[+] and CD8[+] T cells, as observed in IE1 tumors, is predictive for increased T cell expansion upon anti-PD-1 treatment across all breast cancer subtypes[15] and positively influences response to anti-PD-L1 treatment in triple-negative breast cancer[75]. This study also showed that low pre-treatment expression of MHC-I on tumor cells, as observed in IE2 tumors, was associated with reduced T cell expansion upon anti-PD-1 treatment[15]. We hypothesize that immune evasion by MHC-I downregulation, observed in IE2 but not IE1 tumors, may not be amenable to conventional immune checkpoint therapy. Immune evasion of IE1 tumors, on the other hand, is linked to chronic stimulation and a progressively exhausted phenotype of tumor-reactive T cells, and may be targetable by immune checkpoint blockade

In summary, we have described two distinct breast tumor immune environments with respect to tumor antigen presentation, T cell phenotypes, cytotoxic potential, cellular crosstalk, and spatial organization, all of which may affect immunotherapy response. PD-L1 is currently the main marker used to stratify patients for immune checkpoint therapy in breast cancer[76]. Our data suggest that PD-1, CXCL13, and MHC-I, possibly along with previously defined T cell exhaustion markers such as LAG-3 and TIM-3[29], are better able to distinguish immune environments that may prove to be differentially responsive to this therapy. These factors may therefore be useful, together with other patient stratification strategies, to select patients for upcoming clinical trials of immune checkpoint inhibition.

## Methods

### Clinical samples

All tissue samples and health-related data in our study were collected after ethical review and approval of the Cantonal Ethics Committee Zurich (Kantonale Ethikkommission Zürich, #2016-00215) and the faculty of medicine ethics committee at Friedrich-Wilhelms-University Bonn (#255/06).

Primary breast tumor resectates and health-related data were collected in collaboration with the Patient's Tumor Bank of Hope (PATH, Germany) at the breast cancer centers at St. Johannes Hospital Dortmund and Institute of Pathology at Josefshaus (Germany) and the University Hospital Giessen and Marburg, Marburg site (Germany) after obtaining written informed consent from patients. Patients were not compensated. The majority of patient samples sequenced in this study (12 of 14) were originally collected and analyzed by suspension mass cytometry (CyTOF)[29]. For each of these samples, we had access to an aliquot of viably frozen cells derived from the same cell suspension that was used in the original study. These samples were previously classified as either TIG2 or TIG3[29] (corresponding to IE1 and IE2 in this study). This original tumor classification grouped tumors based on shared patterns of infiltrating immune cell cluster frequencies, assessed using the Jensen-Shannon divergence. The remaining two samples (T_BB330 and T_BB338) were collected for this study, and pathological staging was performed by experienced pathologists as previously described[29]. Part of each tumor was formalin-fixed and paraffin-

embedded for the use in standard diagnostics and for analysis by IMC. For CyTOF and scRNA-seq analyses, a tissue sample of about 5 × 5 × 5 mm was taken prior to paraffin embedding. Thus, the tumor area used for standard diagnostics and IMC was spatially separate from the tumor area used for mass cytometry and scRNA-seq. For two out of 14 samples, IMC measurements were not possible due to unsuitable FFPE material or missing patient consent for FFPE-based analysis. Tumor subtypes in this study were defined as follows: Luminal A (ER$^+$ and/or PR$^+$, HER2$^-$, Ki-67$^+$ <20%), Luminal B (ER$^+$ and/or PR$^+$, HER2$^-$, Ki-67$^+$ ≥20%), Luminal B-HER2$^+$ (ER$^+$ and/or PR$^+$, HER2$^+$), HER2$^+$ (ER$^-$, PR$^-$, HER2$^+$), and triple negative (ER$^-$, PR$^-$, HER2$^-$). Two out of 14 patients had received neoadjuvant chemotherapy prior to tumor resection (Supplementary Data 1), but we did not observe any significant difference between tumors from treated and untreated patients in terms of immune cell type frequency or phenotypes. In the original patient cohort, neoadjuvant treated patients were evenly split between the three major TIGs[29].

## Tissue preparation

After surgical resection, fresh tissue samples were immediately transferred to pre-cooled MACS Tissue Storage Solution (Miltenyi Biotec) and were shipped at 4 °C. For suspension mass cytometry, the tissue was processed as previously described[29]. In brief, the Tumor Dissociation Kit, human and the gentleMACS Dissociator (both Miltenyi Biotech) were used for tissue dissociation, followed by filtering of the single-cell suspension. Cells were stained for viability with 25 mM cisplatin (Enzo Life Sciences) and subsequently fixed with 1.6% paraformaldehyde (Electron Microscopy Sciences) before storage at −80 °C. For scRNA-seq, an aliquot of the single-cell suspension was taken prior to viability staining and fixation and viable cells were frozen in RPMI-1640 cell culture medium (Thermo Fisher) supplemented with 10% fetal bovine serum and 10% DMSO (Sigma-Aldrich) using a Mr. Frosty Freezing Container (Thermo Fisher). Viable cell aliquots were stored in liquid nitrogen.

## Suspension mass cytometry

For the two samples not previously analyzed by suspension mass cytometry, we performed mass tag cellular barcoding, antibody labeling and mass cytometry acquisition as previously described using the same antibody clones (immune-centered antibody panel only)[29]. Basic data analysis including dimensionality reduction and clustering were performed as described and allowed allocation of each sample to one of the two distinct immune environments explored in this study (based on the frequency of exhausted T cell phenotypes).

## scRNA-seq sample preparation, data acquisition and pre-processing

For scRNA-seq, viable cell suspensions were quick-thawed in a 37 °C waterbath for <1 min and progressively diluted by slow stepwise addition of RPMI + 10% fetal bovine serum until the volume was ten times that of the initial volume. After a 10-min centrifugation step (300 × g, 4 °C), the supernatant was removed and the Dead Cell Removal Kit (Miltenyi Biotech) was used according to manufacturer's instructions to exclude dead and apoptotic cells. The resulting live cell fraction was washed once with 0.04% bovine serum albumin (BSA) in PBS (Sigma-Aldrich) and resuspended in the same buffer. Cell quality and cell count were assessed using a Neubauer chamber (Electron Microscopy Sciences), and the concentration was adjusted to 800–1000 cells/μl. scRNA-seq libraries were generated using the Chromium Single Cell 3′ Library & Gel Bead Kit v3 from 10x Genomics, aiming for 10,000 single cells per library. All libraries were sequenced on a NovaSEQ6000 System (Illumina). Following quality control, raw sequencing reads were aligned to the human reference genome GRCh38 and reads per gene per cell were quantified using CellRanger (10x Genomics, v3.0.1) (Supplementary Data 2).

## scRNA-seq clustering and cell type annotation

The resulting gene-by-cell matrices from CellRanger were transformed into Seurat objects (Seurat v3.0.2)[77], and downstream analysis was conducted in R v3.6.1 unless otherwise stated. High-confidence doublets were removed using the DoubletFinder package with the pK parameter optimized for each sample individually[78]. Subsequently, all Seurat objects were merged and filtered to exclude cells with >7500 or <200 expressed genes, with >75,000 read counts or with >20% of reads mapping to mitochondrial RNA. The remaining cells were normalized and scaled using the *sctransform* wrapper in Seurat[79]. *Sctransform* also identifies highly variable genes, which were then used to construct Principal Components (PCs). PCs covering the highest variance in the dataset were selected based on elbow plots and heatmaps and used as input for graph-based clustering. Clusters were calculated using the FindNeighbours and FindClusters functions with the resolution parameter set to 2 and visualized using the Seurat implementation of the dimensional reduction algorithm UMAP[80]. Differential gene expression analysis was performed for the 61 resulting clusters using the Seurat implementation of the statistical framework MAST[81]. The main cell types were annotated based on the expression of established marker genes (*EPCAM/CDH1* for epithelial cells, *CD3/CD4/CD8/NCR1* for the T and NK cell fraction, *CD14/ITGAX/HLA-DRA* for myeloid cells, *PECAM1/VWF* for endothelial cells, *PDGFRB/FAP* for fibroblasts, *MS4A2* for mast cells and basophils, *MS4A1* for B cells, and Ig-encoding transcripts for plasma cells) (Supplementary Data 3). Due to large transcriptional overlaps, T and NK cells could not be clearly distinguished at this level and were thus annotated as one cell type. No large-scale batch effects derived from individual patients, sequencing runs or immune environments were found upon visual inspection of the UMAP. Separate clustering of the epithelial cell fractions from individual patient samples is biologically expected and not a batch effect.

For detailed analysis of immune cell subtypes, 14 clusters identified as T and NK cells and 10 clusters identified as myeloid cells were separately subset from the full dataset. Both subsets were re-normalized using *sctransform* while regressing out percentages of mitochondrial RNA and contaminating ambient RNA (specifically, percentages of keratin-encoding transcripts and percentage of *MGP* likely derived from apoptotic/ruptured epithelial cells) as confounding factors. As for the full dataset, PCs covering the highest variance in each subset were selected and used for graph-based subclustering, which was performed at a range of resolutions between 0.1 and 1.5. The final resolution parameter resulting in stable subclusters at desired granularity was selected using clustering tree analysis[82] and was 1 for the T and NK cell subset (Supplementary Fig. 7A) and 0.8 for the myeloid subset. Low-quality subclusters (one T and NK cell cluster with high keratin counts, one T and NK cell cluster and one myeloid cluster with very low read counts, and one myeloid cluster with high mitochondrial percentage) were removed before performing DE analysis. The most differentially expressed genes for each subcluster were used for manual cell subtype annotation (Supplementary Data 3). Due to ambiguous *CD4/CD8* classification of some T cell clusters, we annotated *CD8$^+$*, *CD4$^+$*, or mixed T cell clusters based on the ratio of *CD8A and CD8B* to *CD4* transcript counts (Supplementary Fig. 7E).

## Pseudobulk analysis of scRNA-seq data

For pseudobulk comparison of IE1- versus IE-2-classified samples, raw gene counts were summed over all cells per sample and normalized by library size for boxplot visualization. This was done separately for the grouped T and NK cells, for the myeloid cells, and for the epithelial cells. Differentially expressed genes were identified using the exactTest for single-factor experiments from the edgeR package (v3.26.5)[83]. For this, non-normalized gene-sum matrices were used as input, lowly expressed genes were filtered out (cut-off: >30 read counts in at least

3 samples) and library normalization was performed using the calc-NormFactors function.

To account for tumor grade, a similar analysis was performed with tumor grade as a blocking factor in the model design. Due to the inability of the exactTest to handle complex experimental designs, a quasi-likelihood negative binomial generalized log-linear model (glmQLFTest) was used to determine differentially expressed genes for this analysis[83].

Genes with log counts per million <1.5 in the higher-expressing sample group were excluded from further analysis (Supplementary Data 5).

### Pseudotime analysis of scRNA-seq data

Linear pseudotime ordering was performed for a subset of six T cell clusters (T-naive, T-cytotoxic-1 to -4, and T-CD8-exhausted) using the Bayesian latent variable statistical framework Ouija[47] with 6000 iterations. To speed up calculations, only a random subsample of 800 cells per patient from this T cell subset was used as input. A set of 23 putative marker genes was used for pseudotime learning (Supplementary Data 6) and individual cells were ranked by assigned pseudotime values for further analysis. Mean pseudotimes for patient samples and clusters were calculated as the average pseudotime rank for all T cells of the respective sample/cluster that were included in Ouija analysis.

To study the origin of migDCs, pseudotime analysis without pre-defined topology was performed on a subset of three myeloid clusters (cDC1, cDC2, and migDC) using Slingshot (v1.7.3)[84] and Monocle (v2.12)[85]. Slingshot pseudotime values were extracted for the cDC2-migDC transition (excluding cDC1), cells were ranked by pseudotime and gene expression along pseudotime rank was plotted using the rolling average over 11 cells (selected as a good granularity for visualizing the major trends in the plot, Fig. 4h).

### Identification of myeloid cell gene signatures

In order to interpret gene expression patterns observed across myeloid cells in the context of defined functional states, we compiled four distinct gene signatures. To assemble the T cell-attractive and the T cell-suppressive myeloid cell gene signatures we performed a broad literature survey and manually extracted genes reportedly involved in chemoattraction or myeloid-mediated suppression of T cells (Supplementary Data 6). Gene lists for the M1 and M2 signatures were adopted from Azizi et al.[27]. Prior to summing up signature gene counts to calculate a score for each signature in each cell, individual gene counts were normalized by the total cellular gene count and log-transformed to reduce the relative dominance of highly expressed genes. Given that we only observed positive signature score correlations in our analysis, we also assembled a random control gene signature and found no correlation with either the attractive or the suppressive T cell signature indicating that there were no experimental or computational biases.

### Combined ligand-receptor analysis using scRNA-seq data

**SingleCellSignalR.** To assess cell-to-cell communication, we first used the SingleCellSignalR package[58]. For SingleCellSignalR analysis of main cell type interactions, we subsampled 10,000 cells for each of the cell types and calculated interaction scores for all LR pairs over all cell type pairs. Autocrine and paracrine interaction scores were calculated separately and then merged into a single matrix. An empirically selected LR score threshold of 0.4 was used to define and quantify "true" interaction pairs for each cell type pair as recommended[60] (Fig. 5a and Supplementary Fig. 12A). In order to extract the most relevant hits, we wanted to identify high-scoring, cell type pair-specific interactions. To this end, the 100 highest-scoring LR interactions for each cell type were extracted, followed by identification of the five LR pairs with the highest coefficient of variation between all cell type pairs (i.e.,

the five most specific interactions for the respective cell type pair) (Supplementary Data 7). Of these, pairs were manually selected for plotting based on solidity of the literature evidence and biological interpretability of the interaction (Supplementary Fig. 12C). Single-CellSignalR analysis of the T and NK cell metacluster and myeloid cell metacluster interactions was performed in the same way and with the same cut-offs (Fig. 5c). In Supplementary Fig. 12E, all interactions involving LC, pDC, or T-mixed as well as myeloid-myeloid and T and NK-T and NK interactions were excluded for clarity.

To compare LR interactions between IE1- and IE2-classified tumors, SingleCellSignalR analysis for epithelial cell, myeloid cell, and T and NK cell interactions was performed for each patient sample separately. The difference between LR scores of IE1 and IE2 samples were assessed using Wilcoxon rank-sum testing for each LR pair in each cell type pair. Among all interactions with $p < 0.05$ (not corrected for multiple hypothesis testing) and a mean LR score >0.4, the biologically most relevant and literature-backed interactions were chosen for plotting (Fig. 5b). Complete results can be found in Supplementary Data 6.

**CellPhoneDB.** CellPhoneDB, another algorithm for LR analysis[86], was used to assess the consistency of our SingleCellSignalR results. One major difference between the two algorithms is that CellPhoneDB defines the significance of LR interactions based on random permutation of cluster labels, leading to large $p$ values for interactions that are present in many different cluster pairs. To analyze and quantify the interactions in the main cell types, we subsampled 3000 cells per cell type and ran CellPhoneDB in Python with 1000 iterations (Supplementary Data 8). As for SingleCellSignalR analysis, we compared LR interactions between IE1 and IE2 tumors by running CellPhoneDB on each patient sample separately and testing for enrichment of LR pairs in either of the two IEs. The results were in good agreement with the SingleCellSignalR findings (Supplementary Fig. 12D).

**NicheNet.** To predict which myeloid-derived ligands are the most closely linked to T cell exhaustion in our dataset we used NicheNet, a method that combines single cell expression data with prior knowledge on signaling networks and predicts which sender cell ligands are most likely to have affected the expression of a set of known target genes in the receiver cells[87]. For this analysis, we defined all myeloid cells as sender cells and all CD8+ and naïve T cell clusters as receiver cells. The target gene set included all genes that were specifically upregulated in the T-CD8-exhausted cluster as determined by differential expression analysis. NicheNet quantifies ligand activity by determining the Pearson correlation coefficient between a ligand's a priori target predictions and the observed transcriptional response, with higher ranking ligands having a better ability to predict the target gene set compared to the background of expressed genes.

### Dataset integration

We integrated our scRNA-seq dataset with datasets from two previous studies (Qian and Bassez)[15,57] using integration anchors as implemented in Seurat[77]. Dataset integration and subsequent downstream analysis followed the standard Seurat workflow and was performed separately for immune cells (20,000 cells subset from each dataset) and epithelial cells. From the Bassez[15] dataset, only pre-treatment samples from cohort 1 were included.

### Classification of CD8+ T cell infiltration status

The CD8+ T cell infiltration status was independently assessed by a clinical pathologist based on the spatial distribution of CD8+ T cells on whole-slide H&E and IHC stains as described previously[44]. The samples used for this analysis ($n = 25$) included the 12 samples used in the rest of the study, and 13 additional samples from the cohort in our previous study[29]. In brief, tumors were classified as (1) immune desert if there

 

were no (or very few) CD8+ T cells in the intratumoral and stromal compartment or at the tumor margin, as (2) immune-excluded if CD8+ T cells were only found at the tumor margin or in the stromal compartment but not infiltrating the tumor mass, or as (3) inflamed if CD8+ T cells had infiltrated the tumor mass and were found in direct contact with tumor cells. Desert and immune-excluded phenotypes are generally classified as "immune-cold", while inflamed phenotypes are classified as "immune-hot".

## IMC panels

For spatial analysis of the breast immune environment, we designed and validated two IMC panels that included antibodies against canonical cell type markers and known immunoregulatory targets with a focus on markers differentially expressed in IE1 versus IE2 in our scRNA-seq pseudobulk analysis. The Protein Panel was an all-antibody panel, and the RNA Panel was an RNAscope panel that included 10 cytokine/chemokine-targeting mRNA probes, one negative control mRNA probe (*DapB*), and one mRNA probe targeting *MS4A1* (the gene encoding CD20). We used commercial RNA probes from the RNA-Scope HiPlex v2 kit from Advanced Cell Diagnostics. Details about panels and antibodies is in Supplementary Data 4. All antibodies were validated by immunofluorescence prior to isotope-polymer conjugation, and by IMC after conjugation. Antibody validation included staining of known marker-positive and marker-negative tissue types (including tonsil, lymph node, spleen and different tumor types), assessment of staining pattern across the tissue compared to staining patterns published in the Human Protein Atlas (proteinatlas.org), cell type specificity (assessed by co-staining with other markers) and intercellular location.

## IMC sample preparation and immunofluorescence whole-slide scans

Twelve samples were analyzed by IMC. For each, two consecutive 4-µm thick sections (for staining with two different panels) were cut from the respective FFPE tissue block and processed on slides with a combination of immunofluorescence and IMC staining. Slides intended for staining with the Protein Panel were deparaffinized in HistoClear (Biosystems) three times (10 min per incubation) before being rehydrated in a graded alcohol series (ethanol:deionized water 100:0, 90:10, 80:20, 70:30, 50:50, 0:100; 5–10 min each). Antigen retrieval was performed in Tris-EDTA buffer (pH 9) at 95 °C for 30 min in a NxGen decloaking chamber (Biocare Medical). After cool-down, slides were blocked with 3% BSA in TBS (20 mM Tris (pH 7.6), 150 mM NaCl) for 1 h. Samples were first stained with rabbit anti-CD3 and mouse anti-CD20 overnight at 4 °C. Primary antibodies used at this stage were metal-tagged. Slides were washed three time (5 min per wash) in TBS before staining with fluorescently labeled secondary antibodies (anti-rabbit-Alexa750 and anti-mouse-Alexa555) for 1 h at room temperature. Slides were then stained with DAPI (1:500 in PBS), washed again three times (5 min per wash) in TBS and stained with the remaining metal-tagged antibodies of the Protein Panel overnight at 4 °C. The next day, slides were washed three times (5 min per wash) in TBS and stained with a final concentration of 0.5 µM Cell ID Intercalator-Ir (Fluidigm) for 5 min at room temperature. Slides were washed again in TBS (three times, 5 min per wash), dipped in deionized water, and dried with pressured air.

Slides to be stained with the RNA Panel were processed as previously described[24,33]. Antigen retrieval and RNA staining were performed with an RNAscope Fluorescence Multiplex Reagent Kit (Advanced Cell Diagnostics) according to manufacturer's instructions. Detection oligonucleotides were metal-conjugated as described[24] and used at a final concentration of 20 nM. After the last wash step of the RNAscope protocol, slides were washed for 3 min in TBS and stained with metal-tagged rabbit anti-CD68, mouse anti-panCK, and mouse anti-E-cadherin overnight at 4 °C. Slides were then processed as described above with fluorescently labeled secondary antibody staining followed by DAPI staining, staining with the RNA Panel, and iridium staining before the final washes and slide drying. Immediately after drying, an Axio Scan Z.1 (Zeiss) was used for multi-channel fluorescence whole-slide scanning of all sections.

## IMC region selection and data acquisition

For IMC, four to six representative ROIs were selected manually for each slide based on the immunofluorescence pre-scan. We selected ROIs to be representative of the whole tumor in terms of T cell, macrophage and tumor cell content (i.e., the proportions of these cell types). Immature and mature TLS regions were defined on the immunofluorescence pre-scans as sites of B cell accumulation without a clear center or sites with dense round B cell accumulation, respectively[32]. If present, one to five of these regions were additionally imaged per sample. All ROIs were 1 mm × 1 mm and were manually aligned for each pair of consecutive tissue cuts to ensure the best possible overlap between images from the two different panels. In total, 154 multiplexed images were acquired at 400 Hz using a Hyperion Imaging System (Fluidigm), and the raw data were preprocessed using commercial software (Fluidigm) (Supplementary Data 9).

## IMC data processing

Image files were converted from the commercial format to OME-TIFF, and single cells were segmented using a combination of ilastik[88] v.1.3.3 and CellProfiler[89] v.3.1.9 as described in the workflow available at https://github.com/BodenmillerGroup/ImcSegmentationPipeline[90].
Signal spillover between channels was compensated on the single-cell level using the R package CATALYST[91] (v1.8.7), and high-dimensional averaged marker expression as well as spatial features were extracted for each cell using CellProfiler[90]. Cells on the edges of each image were discarded. Single-cell objects extracted from tissue sections represent small tissue slices that potentially contain overlapping cell fragments despite high-quality segmentation. Especially in densely packed regions, the extracted single-cell marker expression may thus include some information originating from neighboring cells. To extract spatial cellular relationships, the circumference of each cell was expanded by 8 pixels (8 µm) and overlapping cells were defined as neighbors.

## IMC analysis workflow and clustering

IMC downstream analysis was conducted in R v.4.0.2. For each of the two panels, the single-cell expression matrix was converted into a SingleCellExperiment object[92] with the corresponding metadata attached, and a filter was applied to exclude cells with an area <8 pixels or >600 pixels. We used different data transformations for different downstream analysis steps as indicated in the figures: Single-cell marker expression was either (1) arcsinh-transformed, scaled, and centered or (2) 01-normalized for each marker using the 99th percentile normalization to account for outliers. A multi-step graph-based clustering approach based on the 01-normalized data was used to identify single-cell phenotypes. For each clustering step, a shared nearest neighbor graph was built using the scran R package (v.1.16.0)[93] followed by Louvain community detection as implemented in igraph (v.1.2.5)[94]. For each of the panels, the first clustering step was performed with a selected set of cell type markers (Supplementary Data 4), and each cluster was annotated as epithelial or non-epithelial based primarily on expression of panCK and E-Cadherin. Non-epithelial cell clusters were pooled and subjected to a second clustering step with similar markers (excluding epithelial markers; see Supplementary Figs. 2F and 5D for markers used) to identify immune and stromal cell types in higher granularity. Annotation of stromal and immune cell subtypes was based on canonical marker expression. For the Protein Panel data only, T cell clusters were subsetted and subjected to a third clustering step with T cell-specific markers such as PD-

1 and ICOS (see Supplementary Fig. 8B for markers used). To identify Ki-67 + T cells, an empirical cut-off of 0.7 was chosen on Ki-67 arcsinh-transformed counts. In addition, epithelial cells from the Protein Panel that were proliferating, hypoxic, or apoptotic were identified using empirically defined cut-offs for Ki-67 expression, Carbonic Anhydrase XI expression, and Cleaved Caspase/Cleaved PARP expression, respectively. The percentage of Ki-67$^+$ epithelial cells in each patient sample as determined by IMC was strongly correlated to the percentage determined by immunohistochemistry in standard clinical diagnostics (Spearman's $\rho = 0.77$, $p = 0.0037$), confirming sensible cut-off selection. We used the scater implementation[95] of the dimensionality reduction algorithm UMAP for two-dimensional visualization of high-dimensional single-cell data. For the full datasets, the algorithm was run on a random subsample of 10,000 cells per patient sample to avoid overcrowding of data points. For data subsets, the algorithm was run on all cells.

### Cytokine expression status
In the RNA Panel dataset, a binary cytokine expression status for each of the cytokines measured on mRNA level was assigned to every cell as previously described[33]. In brief, the difference between the respective cytokine signal and the signal of the negative control mRNA probe (*DapB*) was determined for each cell, $p$ values were calculated from the distribution of differences, and a threshold of $p < 0.01$ after Benjamini–Hochberg correction was applied to define cytokine-expressing cells.

### Cytokine patches and milieus
Cytokine patches and cytokine milieus were defined as previously described[33]. In short, cytokine patches were defined as a minimum of three neighboring cells all expressing at least one cytokine (general cytokine patch) or all expressing the same cytokine (cytokine-specific patch). For this analysis, the maximum distance for two cells to be considered neighbors was 25 µm. To define cytokine milieus, the cytokine patch borders were extended by 30 µm and all cells within this radius were included in the respective cytokine milieu. For each individual image, enrichment or depletion of a cell type X in a milieu type Y was calculated using a Fisher's exact test. The threshold for significance was $p < 0.01$ and only images that contained both X and Y were used to quantify the percentage of images with significant enrichment or depletion of X in Y.

### Mixed effect models
To account for the presence of multiple images (ROIs) per patient in the dataset, we used mixed effect models wherever possible to compare features between the two IEs. Mixed effect models were estimated and $p$ values calculated using the *mixed* function of the *afex* package (v1.0-1) using default parameters with IE as fixed effect and ROI ID as random effect. For each case, three models were fitted using the original data, log1p-transformed and sqrt-transformed data, respectively, and model fit was assessed by checking the normal distribution of residuals. In cases where none of the three models reached a good fit, individual ROI values were averaged for each sample and Wilcoxon rank sum test was used instead to compare between the two IEs.

### Spatial gene signature validation by IMC
To spatially validate the T cell-attractive myeloid cell gene signatures, we used the RNA Panel dataset and split myeloid cells into two groups based on whether or not they expressed at least one of the measured T cell-attracting chemokines (*CXCL9, CXCL10, CCL2, CCL17, CCL4, CCL5*). Subsequently, the distance to the closest T cell was compared for chemokine-expressing and non-expressing myeloid cells. Of the genes used to construct the T cell-suppressive myeloid cell gene signature,

only antibodies targeting corresponding proteins IDO1 and PD-L1 were included in our IMC panels. For validation of this signature, we thus split myeloid cells of the protein panel IMC dataset into a "suppressive" group (high expression of PD-L1 and/or IDO1) and a "non-suppressive" group. For every myeloid cell, we then identified the ten closest T cells (excluding $T_{regs}$), calculated the proportion of PD-1$^{high}$ T cells among these, and compared this proportion for suppressive versus non-suppressive myeloid cells.

### Pairwise neighborhood analysis
The Protein Panel dataset was used for neighborhood analysis because it allowed a more fine-grained annotation of cellular subtypes than did the RNA Panel. To identify significantly enriched or depleted pairwise neighbor interactions between cell types, a two-sided permutation-test-based analysis with 1000 permutations per image was performed as described previously[60]. The threshold for significant interactions was $p < 0.01$.

### Image visualization
All pixel and single-cell level images shown in the figures were generated using the cytomapper R package (v.1.0.0)[96].

### Whole-slide immunofluorescence screen for TLS classification
To screen for mature TLS and immature TLS, we used FFPE slides from 13 additional patients included in our previous study cohort[29] that were classified as either IE1 or IE2 (corresponding to the previously defined TIG2 and TIG3[29]). Slide pre-processing and antigen retrieval were performed as described above for slides stained with the Protein Panel. After blocking with 3% BSA in TBS, slides were stained with rat anti-CD3, rabbit anti-CD68, and mouse anti-CD20 overnight at 4 °C (Supplementary Data 4). The next day, slides were washed in TBS (three times, 5 min per wash) and stained with fluorescently labeled secondary antibodies (Supplementary Data 4) for 1 h at room temperature. Slides were washed again in TBS (three times, 5 min per wash), and a coverslip was mounted using a polyvinyl alcohol mounting medium. Fluorescent whole-slide images were acquired with an Axio Scan Z.1. Mature and immature TLS sites were defined as above (see "IMC region selection and data acquisition" section) and annotated in an IE-blinded manner.

### Reporting summary
Further information on research design is available in the Nature Portfolio Reporting Summary linked to this article.

## Data availability
The published read count data and metadata from the Bassez dataset[15] was downloaded from https://lambrechtslab.sites.vib.be/en/single-cell. The published counts matrix and metadata from the Qian[57] breast cancer dataset was downloaded from https://lambrechtslab.sites.vib.be/en/pan-cancer-blueprint-tumour-microenvironment-0. All data generated in this study are publicly available. The raw single cell RNA-seq data generated in this study have been deposited in the ArrayExpress database at EMBL-EBI under accession number E-MTAB-10607. The Imaging Mass Cytometry data generated in this study are available on the Zenodo database at DOI: 10.5281/zenodo.4911135. The remaining data are available within the Article and Supplementary Information.

## Code availability
Code used to generate the figures is available in the following repositories: scRNA-Seq data analysis: https://github.com/BodenmillerGroup/BCexh_scRNAseq. IMC data analysis: https://github.com/BodenmillerGroup/BCexh_IMC. Dataset integration: https://github.com/BodenmillerGroup/BC_RNAseq_integration.

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

## Acknowledgements

We thank patients for sample donations and members of the Bodenmiller lab for critical input. The work (B.B.) was supported by the European Research Council (ERC) under the European Union's Horizon 2020 framework, ERC-2019-CoG: 866074—Precision Motifs, and a SNF Project grant (310030_205007). J.W. is supported by a European Molecular Biology Organization (EMBO) Postdoctoral Fellowship ALTF 599-2021.

## Author contributions

S.T. and B.B. conceived the study. S.T. designed, performed and analyzed all scRNA-seq and IMC experiments. T.A., C.L., M.R., and J.W. collected and processed the tissue samples. B.S. performed and analyzed IHC stainings. S.T. and B.B. performed the biological analysis and interpretation. S.T., N.d.S and B.B. wrote and revised the manuscript with input from all authors.

## Competing interests

B.B. is a co-founder of Navignostics, a precision oncology diagnostics company based on multiplexed tumor imaging. All other authors declare no competing interests.
