## [Peer Review File · Nature Communications]

A comprehensive single-cell map of T cell exhaustion-associated immune environments in human breast cancerReviewers' Comments:

Reviewer #1:

Remarks to the Author:

Tietscher and colleagues present a study evaluating immune microenvironments in luminal breast cancer. Specifically, the team conducted both single cell sequencing and imaging mass spec on 14 tumors that were either classified based on the presence of so-called exhausted PD-1+/CTLA-4+/CD38+ T cells. Both the sequencing and imaging yielded a large amount of data that was then analyzed using various bioinformatics tools. Based on the established dichotomy, the authors concluded that there were differences between the two subsets based on several analyses. Of those findings, the authors propose that PD-1 and CXCL13 on T cells and MHC class I on tumor cells are key biomarkers separating the two types of tissues. Overall, the study methods were innovative and the analyses and volume of data were comprehensive. The impact of the findings looks to be relatively modest however, particularly since both PD-1 and CXCL13 are already known biomarkers in breast cancer and are associated with outcomes.

Strengths include:

1. Innovative assays methods and innovative bioinformatic pipelines.
2. Extensive quality control.
3. High quality figures.

Weaknesses include:

1. Incorrect introduction first paragraph which states that there are no approved immunotherapeutic options for other subsets of breast cancer. Arguably both Trastuzumab and Pertuzumab have both ADCC inducing features as well as vaccinal effects. Thus, the paragraph should be rephrased.
2. There are no functional tests or protein-base tests to support the data. Although this problem is alluded to in the conclusions, this is considered a major deficit.
3. It is not clear how the ROIs for the imaging mass spec were selected and if they were representative of the general tumor.
4. Conclusions are made that are sometimes not supported by the data. For example, it is stated that the T cells that were analyzed were tumor reactive but the lack of functional data precludes such a statement.
5. Several genes were differentially expressed but there was no assays to confirm differential protein levels.
6. Sections are long in some cases, difficult to get through and in some cases containing material seemingly unrelated to the theme of the sections.
7. The conclusions has irrelevant sections or sections which are not clearly linked to the results sections.

Reviewer #2:

Remarks to the Author:

General comments:

Very relevant research question and appropriately framed by the introduction. Timely, addressed with state-of-the art technologies. It leads the way in its attempt to draw relevant biological conclusions from patterns in expression and cellular arrangement, connecting single cell phenotypes with cellular and spatial relationships. In a way that is also where the potential pitfalls lie for this work, as the authors frequently draw "functional" conclusions from merely descriptive data. Nevertheless, in most cases they take care to phrase those conclusions carefully and to support any assumptions with relevant references.

1. Both techniques (scRNAseq and IMC) have their caveats, loss of cells by dissociation (e.g. neutrophils), lack of signal or segmentation errors in IMC. How much do the two techniques agree on the proportionality of cell types of the samples in general? A specific example: T and NK cells

constitute a very large proportion of the cell types in the scRNAseq (Fig 1F). Is this proportion realistic, representative, and confirmed by IMC?

2. Based on PD1 expression in the IMC data one would derive a very different separation of samples into IE1 and IE2 (Fig S5C). Should the IMC samples be regrouped into different IE1 and IE2 based on their proportion of exhausted T cells for the analysis, or why not?

Is this discrepancy due to sampling bias? Do different areas of the tissues show different proportions of exhausted T cells? This would be an important notion: It means that there is high intratumour heterogeneity, and extensive sampling per patient would be required to eventually link such profiles to clinical responses. Will it be the area with highest count for PD1+ T cells that is predictive for response, or just the average of all samples?

3. As you have sampled multiple areas per patient, it is something that you can start to address in the data. How much agreement is there between samples within a patient? Figure S2G conveys some of that information. While that figure is beautiful, ranked as it is by tumour proportion, it would provide more insight into the heterogeneity between the samples and differences between groups, if that data were grouped by IE and then by patient.

4. Following from that, as you have multiple ROIs per patient, a statistical test that incorporates that information would be more appropriate (e.g. for fig S5C), especially with high variability seen between samples of one patient. A linear mixed effects model might be worth considering.

In the Discussion, the authors summarise that IE1 is characterised by an exhausted but also cytolytic phenotype. Can you comment on whether the cytolytic T cells, NK and NKT cells, can physically come in contact with the tumour, based on IMC data? Is there any evidence of local cytolytic activity towards the tumour, e.g. by looking at the apoptosis markers in nearby tumour cells?

Specific comments:

1. Separation in groups IE1 (exhausted) and IE2 (non-exhausted): This being the foundation of the study, please specify in more detail what is the proportion/cut-off of exhausted T cells in each group.

2. Figure 1F:

It is unclear from the legend whether this quantification was based on scRNAseq or IMC data.

3. Fig 2B.

Why this analysis on "bulk" level? What cell subsets are these transcription factors mostly associated with? Is this difference due to higher expression levels per cell, or higher abundance of subset?

4. Fig 2G. In transcriptional T cell profiles (Fig 2G), the highest TCF7 is assigned to naïve T cells. Figure S5B shows the clusters of T and NK cells identified by IMC. Cluster 1 (CD4+ PD1-high) contains highest expression of TCF7 and memory cell markers. Would further subsetting of this IMC cluster 1 separate naïve from memory T cells?

5. Line 255

According to the reference provided, Tfh cells are mainly associated with TLS. Can you confirm this association based on the IMC data? And if they aren't associated with TLS, can they still be called Tfh?

6. From line 322

The authors show previously unreported CSF1 expression in the scRNA data. As this is a novel finding, it would be helpful to get visual confirmation of that finding in the IMC. Show us an image example of that what that expression looks like in NKT cells, to convince us that it is valid to use that as a marker

for NKT cells.

Furthermore, do you see association of these NKT cells with (a specific subset of) myeloid cells to support the suggestion that this expression may provide a link between T cell activity and myeloid cell mediated immune responses, as claimed? With NKT cells representing only a minority of the cells (1-6% of all T and NK cells), how likely is it that these are strongly influencing myeloid cell activation and differentiation? To assess this it is also relevant to know what is contribution of the expression of CSF1 in other cells in the tissue, such as the tumour cells? Morandi et al. (PlosOne 2011) showed that 16/17 of their breast cancer cell lines expressed CSF1.

7. CCL18 is one of the most differentially expressed genes between the groups. What subset of myeloid cell is expressing this? Can you include this in one of the figures, e.g. Fig 4D of S7F. Further in the manuscript, discussing the IMC data (Fig 6F) you highlight a CCL18 community to overlap with myeloid cells, which is hardly surprising if it is expressed highly in the myeloid cells. I am not convinced that this illustration is useful. It would be more interesting if any other cell type not expressing the cytokine is enriched in this CCL18 community, such as the Tregs that were said to be expressing the CCR8.

8. Line 367:

“We were able to functionally validate the T cell attraction signature and a subset of markers within the T cell suppression signature using spatial IMC analysis”

It is not clear how you functionally validated the signatures.

Fig S7G: Please provide the names of the T cell attracting cytokines considered for this plot.

Fig S7I-K: It is not clear whether these present data from scRNAseq or IMC nature. For visualisation purposes, please make sure the data is randomised before plotting, as many IE1 events seem to be on top of the graph, or use smaller dots or subsampling to reduce overlap, or perhaps choose an alternative visualisation like a contour plot to show the difference in data spread between the two groups.

9. Figure 6+S10, TLS

Dendritic cells can often be seen in well-developed TLS. Can you comment on whether these are indeed detectable within the TLS that you classify as mature? They do cluster together in the neighbourhood analysis of Fig S10F, but that is also the case in Fig 6A, and Fig S10D suggests that there are more DCs in images with immature TLS rather than mature TLS.

Without showing data about TLS areas only, instead of considering images containing TLS as a whole, one cannot draw a conclusion as the authors did in their last sentence of the results section:

“In addition, our data suggest that migDC-mediated regulation of T cell activity occurs via direct interactions that take place preferentially in regions of mature and immature TLS ...”

It is not entirely clear to me whether I need to compare Fig S10F to Fig 6A to see the enrichment of cellular neighbours in TLS containing images, or that Fig S10F has already the comparison with and without TLS in itself.

Reviewer #3:

Remarks to the Author:

Tietscher et al have used state-of-the-art techniques and sophisticated analyses to comprehensively profile the immune environments of 14 mainly luminal breast tumors. The pairing of single-cell RNAseq with imaging mass cytometry makes for a rich dataset, and the analyses are comprehensive. The work provides an important resource for understanding the heterogeneous cellular organization

and gene/protein expression properties of breast tumors. Despite this, I find some of the interpretations of the observational data one-sided and over-reaching. A more nuanced description of the work, careful consideration of causal inference (in particular given the small sample size and lack of intervention), and more realistic future vision would make this manuscript an excellent contribution to the field.

Major comments:

1) The tumor samples used in this analysis were chosen based on the expression of coinhibitory receptors and CD38 on the surface of extracted T cells. This led to the use of "exhausted" and "non-exhausted" categories to describe the IE1 and IE2 samples. While I appreciate how lengthy it is to say "tumors containing exhausted-like T cells", I found the current terminology and frequent reference to "exhausted immune environments" distracting, particularly as the molecular characteristics of the environments were not necessarily shared and were the very thing being tested. I would recommend only talking about exhaustion (and strictly speaking "exhaustion-like") with respect to the T cells in these tumors.

2) In the course of their in-depth spatial and gene expression analyses, the authors discover other major differences associated with their tumor categories, some of which are structural and visible to a pathologist, or have a more likely causal role in separating tumor types: e.g. hot versus cold; MHCI downregulation by tumor cells; presence of immature TLS. Indeed, as the authors seem to acknowledge, the tumors may be better classified by the presence/activation of infiltrating tumor-reactive T cells or expression of CXCL13 (which of course may themselves be related). It would be helpful to use this information in framing the conclusions because, although the investigation started with a handful of molecular T cell properties, the results speak to an organizational distinction between tumors that goes beyond the vague "inflammatory immune environment" described in the abstract. I would encourage the authors to speak with an immunologist for advice.

3) The authors state that there is no batch effect in scRNAseq (Fig 1B), but there do appear to be many clusters (granular subclusters within Fig 1D cell types) specific to individual tumors. A granular cluster-by-tumor plot should be included, and if single-tumor subsets occur frequently within immune cells, the authors should test whether applying any batch correction dramatically alters their results.

4) It is unclear to me to what extent the results of pseudobulk comparisons of T/NK and myeloid cells are driven by differential abundance of cell types, differential expression within one or a few cell types, or global differential expression. The authors should parse these effects apart.

5) There is an assumption made in the application of NicheNet that myeloid cell signals are upstream of T cell exhaustion. This is not necessarily true and is untestable in an observational single time-point dataset. This analysis should be caveated appropriately.

6) The final suggestion of stratifying patients based on 3 molecular markers appears over-reaching. To my understanding, it is unclear in the first instance whether stratifying based on PD1 expression would improve results of an anti-PD1 trial in luminal breast cancer. So adding a few more correlated markers seems a strange first step. Even in the situation where stratifying based on PD1 expression had proven beneficial, post-hoc analyses of previous trials would be the likely next step before proposing refinement of the criteria. It would benefit the manuscript to make this forward-looking statement more realistic.

Minor comments:

1) While the authors have tested the impact of tumor grade on their results, the dataset is still composed of a small number of tumors from each of several different types of breast cancer. It would be helpful to acknowledge this heterogeneity and describe if it impacts any results (not a full re-analysis, but looking at any association with clustering and key findings).

2) The sample size in this study is small by necessity. Where appropriate, the potential for spurious correlation should be acknowledged.

3) I am surprised to see strong e-cadherin expression on T/NK cells in Figure S2F. Can the authors explain this?

4) For binary cytokine expression status assigned using the RNA Panel of the IMC data, it would be

helpful to see a plot summarizing these results – how frequent were cytokine+ cells? Which cells expressed which cytokines? At the moment it is difficult to assess how realistic the results from this analysis might be.

5) Line 177: "Expression of T cell exhaustion markers at the protein level is the most distinguishing feature of the two immune environments IE1 and IE2 in our study cohort." Where is the data to support this? Later data in the paper suggests that CXCL13 is actually better.

6) I cannot find a text reference for Figure S4G.

7) Line 246: "HLA-ABC" is not a protein. It refers to HLA-A, HLA-B, and HLA-C, which are all MHC class I proteins.

8) Pseudobulk comparison of 9 selected genes in T cells and NK cells does not seem statistically appropriate. Given the genome-wide nature of the dataset and global perspective of the paper, this analysis should control the genome-wide false discovery rate. The effect sizes can still be discussed, but I would recommend not using a cherry-picked list to assess statistical significance.

9) Call to Figure S4F in line 309 appears wrong.

Reviewer #4:

Remarks to the Author:

In this study, Tietscher et al. attempt to compare transcriptional signatures of exhausted T cells in luminal breast cancer. The focus of the study is of interest to the field, however, we question the novelty of this manuscript as it stands. We would recommend further work to extend the mechanisms and pathways highlighted in this work.

1) First and foremost, much of the most significant differences shown between IE1 and IE2 were circular. The initial classification of tumors was described as follows: "samples contained exhausted T cells (i.e., PD-1^{high}/CTLA-4^{high}/CD38^{high} T cells); we annotated these as immune environment 1 (IE1). The other half mainly contained T cells that did not express exhaustion markers and were annotated as immune environment 2 (IE2)". Therefore, the finding that there are more exhaustion-associated markers such as BATF/TOX and more exhausted/proliferative cells in IE1 is unsurprising and almost definitional. Similarly, the finding in Fig. 4 that TAMs and cDC2s (which are associated with wound-healing) are enriched in immune-suppressed tumors are not surprising.

2) The identification of TLS differences between IE1 and IE2 tumors is potentially interesting, but this part of the manuscript needs to be clarified and strengthened. Would the analyses in Fig6A - D look different between IE1 and IE2 if the tumors were separated into TLS and non-TLS areas? Were migDCs, which seemed to have equal numbers in IE1 vs IE2 tumors by scRNAseq, differentially abundant by spatial analysis? Since CXCL13 "did not differ significantly between images with immature versus mature TLS", are there other possibilities in the cytokine profiles that might be candidates for later experimental validation? Finally, do the authors see any evidence of direct tumor cell/T cell interactions and if so, which T cell types are found most proximal to the tumor cells?

3) The goal of identifying a program of exhaustion in luminal breast cancer is laudable, however it is not clear how these findings compare to other single-cell profiling studies in TNBC or other breast cancer types. It is critical to compare these data to other published datasets, both to confirm the robustness of these findings in other cohorts and also to understand which of these findings might be context-dependent vs. represent pan-breast-cancer biology. Some possible refs that are already cited by the authors and are worth comparing to:

Bassez, A. et al. A single-cell map of intratumoral changes during anti-PD1 treatment of patients with breast cancer. *Nature Medicine* (Springer US, 2021).
doi:10.1038/s41591-021-01323-8.

Azizi, E. et al. Single-Cell Map of Diverse Immune Phenotypes in the Breast Tumor Microenvironment.

Cell 174, 1–16 (2018).

Wagner, J. et al. A Single-Cell Atlas of the Tumor and Immune Ecosystem of Human Breast Cancer. Cell 177, 1–16 (2019).

Qian, J. et al. A pan-cancer blueprint of the heterogeneous tumor microenvironment revealed by single-cell profiling. Cell Res. 30, 745–762 (2020).

Reviewer #1, expertise in breast cancer immunotherapies (Remarks to the Author):

Tietscher and colleagues present a study evaluating immune microenvironments in luminal breast cancer. Specifically, the team conducted both single cell sequencing and imaging mass spec on 14 tumors that were either classified based on the presence of so-called exhausted PD-1+/CTLA-4+/CD38+ T cells. Both the sequencing and imaging yielded a large amount of data that was then analyzed using various bioinformatics tools. Based on the established dichotomy, the authors concluded that there were differences between the two subsets based on several analyses. Of those findings, the authors propose that PD-1 and CXCL13 on T cells and MHC class I on tumor cells are key biomarkers separating the two types of tissues. Overall, the study methods were innovative and the analyses and volume of data were comprehensive. The impact of the findings looks to be relatively modest however, particularly since both PD-1 and CXCL13 are already known biomarkers in breast cancer and are associated with outcomes.

Strengths include:

1. Innovative assays methods and innovative bioinformatic pipelines.
2. Extensive quality control.
3. High quality figures.

We thank the referee for their comments.

Weaknesses include:

1. Incorrect introduction first paragraph which states that there are no approved immunotherapeutic options for other subsets of breast cancer. Arguably both Trastuzumab and Pertuzumab have both ADCC inducing features as well as vaccinal effects. Thus, the paragraph should be rephrased.

Trastuzumab and Pertuzumab both target HER2, but as the referee states, have secondary immune effects. We have rephrased to "there are no approved immunotherapeutic options for patients with luminal HER2-negative breast cancer subtypes".

2. There are no functional tests or protein-base tests to support the data. Although this problem is alluded to in the conclusions, this is considered a major deficit.

The IMC data are protein-based and in part report on markers indicative of function such as chemokines. But the referee is correct that we do not have functional tests such as reporter assays or perturbation experiments in this study. We had already been careful in the initial submission to make clear that functional interpretations were speculative, and we have now reduced such aspects even further. We also mentioned in the discussion that, given the available samples, functional assays were not possible.

3. It is not clear how the ROIs for the imaging mass spec were selected and if they were representative of the general tumor.

The ROIs were selected manually from whole-slide immunofluorescence (IF) scans. We selected them to be representative of the whole tumor in terms of the T cell, macrophage and tumor cell content. We now

describe ROI selection in more detail in the Methods section “IMC region selection and data acquisition”. We show here an example of a whole-slide IF scan with an overlay of selected ROIs (Figure R1).

Figure R1: Whole-slide IF scans of two sections of the same FFPE breast cancer sample (TBB338), stained with the RNA panel (top) and the Protein panel (bottom). Red squares indicate the ROIs that were acquired by IMC.

4. Conclusions are made that are sometimes not supported by the data. For example, it is stated that the T cells that were analyzed were tumor reactive but the lack of functional data precludes such as a statement.

It is correct that, in this retrospective analysis of FFPE patient tissue, we necessarily do not have functional assays and therefore draw conclusions based on an interpretation of marker expression. In the manuscript, we already clearly described which marker signatures we interpreted as indicating tumor reactivity (i.e., 4-1BB, GITR, CXCL13, ENTPD1). We have now further revised to make clearer that by “tumor reactive” we mean that the cells display a marker signature of tumor reactivity.

5. Several genes were differentially expressed but there was no assays to confirm differential protein levels.

We confirmed differential expression at the protein level for some of the most interesting genes (e.g., PD-1, CXCL13, GZMB, MHC-I) using IMC. Panel design for IMC is constrained by antibody availability and also by the need to use several markers to define cell types, without which the data are difficult to interpret.

6. Sections are long in some cases, difficult to get through and in some cases containing material seemingly unrelated to the theme of the sections.

The manuscript describes both a single-cell resource and a highly-multiplex, single-cell, spatial comparison of immune environments in breast cancer. We suspect that the referee’s comments pertain to the resource aspect. We had tried to balance these aspects throughout the manuscript, and based on the referee’s comments, we have shortened the resource description further.

7. The conclusions has irrelevant sections or sections which are not clearly linked to the results sections.

We have tried to link the two sections better in the revised manuscript.

Reviewer #2, expertise in imaging mass cytometry/TME (Remarks to the Author):

General comments:

Very relevant research question and appropriately framed by the introduction. Timely, addressed with state-of-the art technologies. It leads the way in its attempt to draw relevant biological conclusions from patterns in expression and cellular arrangement, connecting single cell phenotypes with cellular and spatial relationships. In a way that is also where the potential pitfalls lie for this work, as the authors frequently draw “functional” conclusions from merely descriptive data. Nevertheless, in most cases they

take care to phrase those conclusions carefully and to support any assumptions with relevant references.

We thank the referee for their encouraging comments. As mentioned above, we have now made even clearer which data support our “functional” interpretations.

1. Both techniques (scRNAseq and IMC) have their caveats, loss of cells by dissociation (e.g. neutrophils), lack of signal or segmentation errors in IMC. How much do the two techniques agree on the proportionality of cell types of the samples in general? A specific example: T and NK cells constitute a very large proportion of the cell types in the scRNAseq (Fig 1F). Is this proportion realistic, representative, and confirmed by IMC?

We show here the proportions of all major cell types as determined by scRNAseq, IMC and suspension mass cytometry (Figure R2). As expected, the proportions do vary somewhat between techniques. The T-NK cells are quite variable between samples, and this is reflected also in mass cytometry measurements. We have added this figure to the revised manuscript as Figure S3A.

Figure R2. Line plot comparing relative cell type frequencies as determined by IMC, CyTOF and scRNA-seq analysis of the same sample. Only samples analyzed with all three technologies are included.

2. Based on PD1 expression in the IMC data one would derive a very different separation of samples into IE1 and IE2 (Fig S5C). Should the IMC samples be regrouped into different IE1 and IE2 based on their proportion of exhausted T cells for the analysis, or why not?

Is this discrepancy due to sampling bias? Do different areas of the tissues show different proportions of exhausted T cells? This would be an important notion: It means that there is high intratumour

heterogeneity, and extensive sampling per patient would be required to eventually link such profiles to clinical responses. Will it be the area with highest count for PD1+ T cells that is predictive for response, or just the average of all samples?

The reviewer is correct that the PD-1 expression in IMC would not group the samples in exactly the same way as it would in the scRNA-seq data. This may be due in part to intratumour heterogeneity, as the reviewer suggests. However, it is likely also affected by the fact that the PD-1 signal in IMC is generally low (i.e., close to the detection limit), which may explain why it is not detected in two of the IE1 samples. We note that CXCL13, which has generally much better signal in IMC, does group the samples in exactly the same way as it does in the scRNA-seq data (Fig. 2K), further supporting this grouping as well as our reasoning for the difference seen with PD-1. Nevertheless, we agree with the reviewer that spatial heterogeneity is an important issue, and we have addressed it further in the next two points.

3. As you have sampled multiple areas per patient, it is something that you can start to address in the data. How much agreement is there between samples within a patient? Figure S2G conveys some of that information. While that figure is beautiful, ranked as it is by tumour proportion, it would provide more insight into the heterogeneity between the samples and differences between groups, if that data were grouped by IE and then by patient.

Thanks for this suggestion. We have now grouped the data in Figure S2G by IE and by patient, as suggested, and have updated the figure in the revised manuscript. It is also reproduced here for convenience (Figure R3). We observe both inter- and intra-patient variation, which is indeed more apparent in the revised plot.

Fig R3: Stacked barplot of IMC protein panel data displaying cell type compositions for all individual images. Patient sample, IE classification, and TLS status are indicated.

4. Following from that, as you have multiple ROIs per patient, a statistical test that incorporates that information would be more appropriate (e.g. for fig S5C), especially with high variability seen between samples of one patient. A linear mixed effects model might be worth considering.

Thanks for this suggestion. We have now used mixed effect models wherever possible (i.e., where the data fit the assumptions of the MEM) when comparing features between the two IEs based on IMC data, to account for the presence of multiple ROIs per patient. This led to revised Fig. 2I (reproduced below in Figure R4), S5C, S5D and S6A. The overall conclusions remain unchanged.

We have updated the Methods section as follows: “Mixed effect models were estimated and p-values calculated using the mixed function of the afex package (v1.0-1) using default parameters with IE as fixed effect and ROI ID as random effect. For each case, three models were fit using either the original data, log_{1p}-transformed and sqrt-transformed data; model fit was assessed by checking the normal distribution of residuals. In cases where none of the three models reached a good fit, individual ROI values were averaged for each sample and Wilcoxon rank sum test was used instead to compare between the two IEs.”

Figure R4. Use of mixed effect model - comparison of data in initial submission versus revised manuscript. Boxplot comparing the mean single-cell HLA-ABC expression in IMC data for the epithelial subsets of IE1 versus IE2 samples. The left plot shows the data averaged over all cells per patient (Figure 2I in initial submission), with each dot representing one patient. The right plot shows the data averaged over all cells per image (Figure 2I, revised manuscript), with each dot representing one image, and images separated by patient. A mixed effects model was fitted on the log_{1p}-transformed data.

In the Discussion, the authors summarise that IE1 is characterised by an exhausted but also cytolytic phenotype. Can you comment on whether the cytolytic T cells, NK and NKT cells, can physically come in contact with the tumour, based on IMC data? Is there any evidence of local cytolytic activity towards the tumour, e.g. by looking at the apoptosis markers in nearby tumour cells?

Using Granzyme B expression as a proxy for a cytolytic phenotype, our IMC data shows that cytolytic T and NK cells do indeed directly contact tumor cells. However, their average number of direct tumor cell neighbors does not differ from those of non-cytolytic (Granzyme B-low) T and NK cells (Figure R5, left). The proportion of T and NK cells with a cytolytic phenotype also did not correlate with the proportion of apoptotic T cells at the image level (Figure R5, right). However, since Granzyme B is certainly not the only cytolytic molecule expressed by T cells, the IMC data is most probably insufficient to fully characterize cytolytic T cell phenotypes. The IMC panel does not include additional cytolytic T cell or apoptosis markers.

Figure R5. Left: Paired box plot comparing the average number of direct tumor cell neighbors for Granzyme B-high and Granzyme B-low T and NK cells. Each pair of dots represents a separate sample. A paired Wilcoxon rank sum test was used for statistical analysis. **Right:** Scatterplot of the proportion of apoptotic tumor cells vs the proportion of Granzyme B-high T and NK cells. Each dot represents one image. T and NK cells were defined as Granzyme-B high if they had an average Granzyme B count > 1.

Specific comments:

1. Separation in groups IE1 (exhausted) and IE2 (non-exhausted): This being the foundation of the study, please specify in more detail what is the proportion/cut-off of exhausted T cells in each group.

For 12 out of our 14 samples, this was previously defined based on suspension mass cytometry data as described in Wagner et al (doi: 10.1016/j.cell.2019.03.005; note that in Wagner et al, IE1 and IE2 were referred to as tumor-immune group or TIG2 and TIG3 respectively). The samples were defined by grouping them according to shared patterns of immune cell phenotypes, assessed using the Jensen-Shannon divergence to measure the similarity between probability distributions representing population frequencies. Thus, the IE1 versus IE2 classification is not based on a simple cut-off of exhausted T cells.

However, once the groups were defined, it was observed that they differed with respect to T cell exhaustion phenotypes.

For the remaining two samples we used additionally in our study, we selected the samples based on similar T cell phenotypic composition (i.e., proportions of PD-1 high T cells and Tregs) to the previously defined IE1 and IE2 samples. Again, for these two samples we did not use a pre-defined T cell cut-off. We have added further explanation to both the results and the methods section to make this clearer.

2. Figure 1F:

It is unclear from the legend whether this quantification was based on scRNAseq or IMC data.

It was based on scRNAseq data, we have now made this clear in the legend.

3. Fig 2B.

Why this analysis on “bulk” level? What cell subsets are these transcription factors mostly associated with? Is this difference due to higher expression levels per cell, or higher abundance of subset?

We did the bulk analysis simply as an initial overview of expression patterns. We then went on to single-cell analysis precisely to tease out the reviewer’s questions. These data are in Figure 2D-2G in the case of T cells and we have now added the single-cell transcription factor expression patterns to revised Figure 2F. The difference between IE1 and IE2 is partly due to higher expression per cell and partly due to higher abundance of particular cell types.

For example, the transcription factor BATF is especially high in Tregs, whose abundance is similar in IE1 and IE2, but within the Treg subset it is higher in IE1 cells compared to IE2 (Figure R6). Additionally, BATF is also high in Tfh cells, which are more abundant in IE1 (Fig 2E in the manuscript).

Fig R6: Violin plot of SCTransformed BATF transcript counts of single Tregs from IE1 vs IE2 samples.

4. Fig 2G. In transcriptional T cell profiles (Fig 2G), the highest TCF7 is assigned to naïve T cells. Figure S5B shows the clusters of T and NK cells identified by IMC. Cluster 1 (CD4+ PD1-high) contains highest expression of TCF7 and memory cell markers. Would further subsetting of this IMC cluster 1 separate naïve from memory T cells?

This is a good point. We have tried clustering the T/NK cells in IMC at different resolutions, but the clusters remain roughly the same, suggesting that subclustering based explicitly on TCF7 is also not likely to yield interpretable cell types. We note that it is also possible that TCF7 gene expression does not correlate well with TCF7 protein expression, as we mention in the text (line 226, Fig S5D). We do not have other memory cell markers in the IMC panel.

5. Line 255

According to the reference provided, Tfh cells are mainly associated with TLS. Can you confirm this association based on the IMC data? And if they aren't associated with TLS, can they still be called Tfh?

Yes, Tfh cells are associated with TLS. This is shown in Fig S10D (see the panel for the CD4+ PD1-high T cells).

6. From line 322

The authors show previously unreported CSF1 expression in the scRNA data. As this is a novel finding, it would be helpful to get visual confirmation of that finding in the IMC. Show us an image example of that what that expression looks like in NKT cells, to convince us that it is valid to use that as a marker for NKT cells.

CSF1 staining is shown in the newly added Fig S13, top right, and reproduced here (Figure R7). We note that there are no defined NKT markers. We defined NKT cells as any cell previously defined as an NK or T cell that was also CSF-1 positive.

Figure R7. Exemplary IMC image of breast cancer sample from our cohort, showing staining pattern of anti-CSF1. CD7 is a marker for T and NK cells, HH3 is a nuclear marker, CXCL9 is another cytokine and DapB is the mRNA negative control.

Furthermore, do you see association of these NKT cells with (a specific subset of) myeloid cells to support the suggestion that this expression may provide a link between T cell activity and myeloid cell mediated immune responses, as claimed?

We do observe a spatial association of CSF-1 expressing NKT cells with activated myeloid cells (defined as myeloid cells expressing at least one cytokine-encoding mRNA), but this is true also for T and NK cells expressing most other cytokines, so the observation is not specific to CSF-1 expressing cells (Figure R8).

Figure R8: Smoothed line plot of the proportion of myeloid cells expressing at least one cytokine-encoding mRNAs versus distance to the nearest T/NK/NKT cell expressing the indicated cytokines. Individual cells were binned in 5- μm steps. Myeloid cells from images with no T/NK/NKT cells or with a distance to these cells > 800 μm were assigned to the maximum bin of 800 μm .

With NKT cells representing only a minority of the cells (1-6% of all T and NK cells), how likely is it that these are strongly influencing myeloid cell activation and differentiation? To assess this it is also relevant to know what is contribution of the expression of CSF1 in other cells in the tissue, such as the tumour cells? Morandi et al. (PlosOne 2011) showed that 16/17 of their breast cancer cell lines expressed CSF1.

The referee is correct that CSF-1 is likely to be expressed by other cells in the tissue. We do not claim that NKT cells are the only source of CSF-1.

7. CCL18 is one of the most differentially expressed genes between the groups. What subset of myeloid cell is expressing this? Can you include this in one of the figures, e.g. Fig 4D of S7F.

We have added CCL18 to Fig S7F (reproduced here as Figure R9). This shows that the CCL18 transcript is exclusively expressed in TAMs, being highest in TAM-7, TAM-4 and TAM-5.

Figure R9: Heatmap showing normalized average expression of myeloid markers commonly used for subtype classification in FACS and CyTOF experiments.

Further in the manuscript, discussing the IMC data (Fig 6F) you highlight a CCL18 community to overlap with myeloid cells, which is hardly surprising if it is expressed highly in the myeloid cells. I am not convinced that this illustration is useful. It would be more interesting if any other cell type not expressing the cytokine is enriched in this CCL18 community, such as the Tregs that were said to be expressing the CCR8.

We have made clear in the revised manuscript that it is expected to find myeloid cells enriched in CCL18 milieu. We unfortunately cannot assess Treg enrichment directly in the CCL18 community since we have no Treg marker in our RNA panel. However, CD4+ T cells are the second-most enriched cell type in the CCL18 community after myeloid cells, which may be due to Treg enrichment, and would be consistent with a CCR8/CCL18 interaction.

8. Line 367:

“We were able to functionally validate the T cell attraction signature and a subset of markers within the T cell suppression signature using spatial IMC analysis”

It is not clear how you functionally validated the signatures

Yes, this wording was not precise. We mean that myeloid cells that express cytokines within the T-cell attracting signature are spatially closer to T cells than myeloid cells that do not express these cytokines; and that T cells in the vicinity of myeloid cells that express one of the suppressive markers PD-L1 or IDO1 more often display an exhausted phenotype compared to T cells in the vicinity of other myeloid cells (Figure S7G and H). To keep the text concise, we have rephrased as “spatial IMC analysis further

supported the T cell attraction signature and a subset of markers within the T cell suppression signature”.

Fig S7G: Please provide the names of the T cell attracting cytokines considered for this plot.

The cytokines that were considered are (CXCL9, CXCL10, CCL2, CCL17, CCL4, CCL5). We have now added these to Fig S7G.

Fig S7I-K: It is not clear whether these present data from scRNAseq or IMC nature. For visualisation purposes, please make sure the data is randomised before plotting, as many IE1 events seem to be on top of the graph, or use smaller dots or subsampling to reduce overlap, or perhaps choose an alternative visualisation like a contour plot to show the difference in data spread between the two groups.

The signatures in Figures S7I-K are defined from scRNA-Seq data. We have now made this clear in the figure legend. We have also adjusted Figure S7I-K as suggested (reproduced here as Figure R10), by randomising the data before plotting and decreasing the opacity and the size of the single points.

Figure R10. Reproduction of revised Figures S7I-K for referee. Left (revised Figure S7I): Single-cell scatterplot of the T cell-attraction score versus the T cell-suppression score. Scores are defined from scRNASeq data. Right (revised Figure S7K): Single-cell scatterplot of the M1 score versus the M2 score. Scores are defined from scRNASeq data.

9. Figure 6+S10, TLS

Dendritic cells can often be seen in well-developed TLS. Can you comment on whether these are indeed detectable within the TLS that you classify as mature? They do cluster together in the neighbourhood analysis of Fig S10F, but that is also the case in Fig 6A, and Fig S10D suggests that there are more DCs in images with immature TLS rather than mature TLS.

Yes, dendritic cells are detectable within mature TLS. Note that, other than the migDC and pDC subsets, DCs are simply classified as 'myeloid' cells in our IMC data. All of these subsets (myeloid cells, migDCs and pDCs) are detectable in mature TLS.

As the referee points out, migDCs and pDCs show a trend toward higher numbers in immature versus mature TLS, but these differences are not statistically significant.

Without showing data about TLS areas only, instead of considering images containing TLS as a whole, one cannot draw a conclusion as the authors did in their last sentence of the results section: "In addition, our data suggest that migDC-mediated regulation of T cell activity occurs via direct interactions that take place preferentially in regions of mature and immature TLS ..."

We propose that the whole image/ROI containing a TLS could be considered a 'region of mature or immature TLS' since the TLS typically occupies 10-25% of the ROI area. Also, in particular for immature TLS it is very difficult to clearly draw a border between this structure and its surrounding. Nevertheless, we have rephrased the text to now say "in regions at or near mature and immature TLS".

It is not entirely clear to me whether I need to compare Fig S10F to Fig 6A to see the enrichment of cellular neighbours in TLS containing images, or that Fig S10F has already the comparison with and without TLS in itself.

Yes, one would need to compare Fig S10F to Fig 6A to see this. We have now mentioned this when we describe the result.

Reviewer #3, expertise in sc-RNAseq/TME (Remarks to the Author):

Tietscher et al have used state-of-the-art techniques and sophisticated analyses to comprehensively profile the immune environments of 14 mainly luminal breast tumors. The pairing of single-cell RNAseq with imaging mass cytometry makes for a rich dataset, and the analyses are comprehensive. The work provides an important resource for understanding the heterogeneous cellular organization and gene/protein expression properties of breast tumors. Despite this, I find some of the interpretations of the observational data one-sided and over-reaching. A more nuanced description of the work, careful consideration of causal inference (in particular given the small sample size and lack of intervention), and more realistic future vision would make this manuscript an excellent contribution to the field.

Major comments:

1) The tumor samples used in this analysis were chosen based on the expression of coinhibitory receptors and CD38 on the surface of extracted T cells. This led to the use of "exhausted" and "non-exhausted" categories to describe the IE1 and IE2 samples. While I appreciate how lengthy it is to say "tumors containing exhausted-like T cells", I found the current terminology and frequent reference to "exhausted immune environments" distracting, particularly as the molecular characteristics of the environments were not necessarily shared and were the very thing being tested. I would recommend

only talking about exhaustion (and strictly speaking “exhaustion-like”) with respect to the T cells in these tumors.

This is a good point. In our initial submission we had already attempted to describe these environments both concisely and accurately, but we have further revised this based on the referee’s comments. For reasons of brevity, we do not call IE1/2 “tumors containing/not containing exhausted-like T cells”. However, we have made the following changes: we now refrain from referring to exhausted/non-exhausted immune environments until clearly defining what these mean in the introduction and at the beginning of the results. We have removed the terms exhausted and non-exhausted IEs in the abstract. We have refined the text in several concluding statements to be clearer that these are immune environments either with or without signs of T cell exhaustion, rather than referring to them as exhausted/non-exhausted. We hope this now strikes a better balance.

2) In the course of their in-depth spatial and gene expression analyses, the authors discover other major differences associated with their tumor categories, some of which are structural and visible to a pathologist, or have a more likely causal role in separating tumor types: e.g. hot versus cold; MHC1 downregulation by tumor cells; presence of immature TLS. Indeed, as the authors seem to acknowledge, the tumors may be better classified by the presence/activation of infiltrating tumor-reactive T cells or expression of CXCL13 (which of course may themselves be related). It would be helpful to use this information in framing the conclusions because, although the investigation started with a handful of molecular T cell properties, the results speak to an organizational distinction between tumors that goes beyond the vague “inflammatory immune environment” described in the abstract. I would encourage the authors to speak with an immunologist for advice.

Thank you for your suggestion. We have revised the discussion with these points in mind.

3) The authors state that there is no batch effect in scRNAseq (Fig 1B), but there do appear to be many clusters (granular subclusters within Fig 1D cell types) specific to individual tumors. A granular cluster-by-tumor plot should be included, and if single-tumor subsets occur frequently within immune cells, the authors should test whether applying any batch correction dramatically alters their results.

A cluster-by-tumor plot is included for T cell sub-clusters (Fig S4C) and myeloid cell subclusters (Fig S7E); these are also reproduced here for convenience (Figure R11). There are indeed a few patient-specific clusters, but no obvious batch effect since most clusters are present in all patients.

We now also provide a cluster-by-tumor plot for the initial (full-dataset) clustering (Figure R12), but have not included it in the manuscript as the T cell and myeloid cell subclusters are the focus of our study.

Figure R11. Cluster-by-tumor plots for T cell and myeloid cell clusters (Fig S4C and Fig S7E). Stacked barplots showing the number of cells originating from each patient sample in each of the scRNA-seq T cell clusters (top) and myeloid clusters (bottom).

Figure R12. Cluster-by-tumor plot for all clusters in the dataset. Stacked barplots showing the proportion of cells originating from each patient sample in each cluster (scRNA-seq dataset).

4) It is unclear to me to what extent the results of pseudobulk comparisons of T/NK and myeloid cells are driven by differential abundance of cell types, differential expression within one or a few cell types, or global differential expression. The authors should parse these effects apart.

We included the bulk analyses simply as an initial overview of general expression patterns. The following single-cell analyses then tease out whether these differences are due to differential abundance of cell types or differential expression within one or more cell types. These data are in Figure 2D-2G in the case of T cells. The difference between IE1 and IE2 is partly due to higher expression per cell and partly due to higher abundance of particular cell types.

For example, and as also stated in our response to reviewer 2, the transcription factor BATF is especially high in Tregs, whose abundance is similar in IE1 and IE2, but within the Treg subset it is higher in IE1 cells compared to IE2 (Figure R6 above). Additionally, BATF is also high in Tfh cells, which are more abundant in IE1 (Fig 2E in the manuscript).

5) There is an assumption made in the application of NicheNet that myeloid cell signals are upstream of T cell exhaustion. This is not necessarily true and is untestable in an observational single time-point dataset. This analysis should be caveated appropriately.

This is a good point. We have added the following sentence to the text describing the NicheNet analysis "We cannot rule out that myeloid cells are not only upstream but also downstream of exhausted T cells."

6) The final suggestion of stratifying patients based on 3 molecular markers appears over-reaching. To my understanding, it is unclear in the first instance whether stratifying based on PD1 expression would improve results of an anti-PD1 trial in luminal breast cancer. So adding a few more correlated markers seems a strange first step. Even in the situation where stratifying based on PD1 expression had proven beneficial, post-hoc analyses of previous trials would be the likely next step before proposing refinement of the criteria. It would benefit the manuscript to make this forward-looking statement more realistic.

We have softened this statement, and acknowledge (i) that it depends on linking IEs to outcome and (ii) that there are other such strategies.

Minor comments:

1) While the authors have tested the impact of tumor grade on their results, the dataset is still composed of a small number of tumors from each of several different types of breast cancer. It would be helpful to acknowledge this heterogeneity and describe if it impacts any results (not a full re-analysis, but looking at any association with clustering and key findings).

Our samples are mostly luminal. There is only one sample each of the TN and LumB/Her2+ subtypes, so we did not include any further analysis of associations with molecular subtypes.

While we could include a further analysis of LumA vs LumB tumors, we decided not to do so since our samples are selected to be immune-high, such that they are not entirely representative of luminal tumors generally. Note that we explicitly excluded lumA tumors from the original cohort that are immune-excluded and have now made this clearer in the revised manuscript.

We have therefore addressed the referee's concern by further emphasizing in the discussion section: the nature of the samples we tested, the relatively small number of tumors, and the heterogeneity in terms of grade and subtype.

2) The sample size in this study is small by necessity. Where appropriate, the potential for spurious correlation should be acknowledged.

We have now acknowledged the general potential for spurious correlation due to small sample size in the discussion. We have also specifically acknowledged the potential for a spurious relationship between NKT cell frequencies and T cell exhaustion.

3) I am surprised to see strong e-cadherin expression on T/NK cells in Figure S2F. Can the authors explain this?

The single T/NK cluster that shows e-cadherin expression likely consists of cells that border tumor cells and thus show some artifactual signal due to imperfect segmentation; this is unfortunately unavoidable in dense tissues and with the current resolution of IMC. Note that this signal is much lower than that of a tumor cell - the reason it appeared strong in this case is because the signal is row-normalized in the heatmap and tumor cells are not included in the plot. As a consequence, the stromal/immune cluster with the highest E-cadherin expression is assigned the value 1. To avoid confusion, we have removed E-cadherin from the heatmap.

4) For binary cytokine expression status assigned using the RNA Panel of the IMC data, it would be helpful to see a plot summarizing these results – how frequent were cytokine+ cells? Which cells expressed which cytokines? At the moment it is difficult to assess how realistic the results from this analysis might be.

We now show the numbers of cells of different cell types expressing various cytokines in Figure S3F, and reproduce it here for the reviewer (Figure R13).

Figure R13. Reproduction of revised Figure S3F. Stacked barplot showing the frequency of positive cells for each of the measured cytokine mRNAs, colored by cell type.

5) Line 177: “Expression of T cell exhaustion markers at the protein level is the most distinguishing feature of the two immune environments IE1 and IE2 in our study cohort.” Where is the data to support this? Later data in the paper suggests that CXCL13 is actually better.

This sentence refers to the features that originally distinguished IE1 and IE2 tumors in mass cytometry data (Wagner et al). We have rephrased the text, now stating “The two immune environments in our study cohort were defined based on protein-level expression of T cell exhaustion markers (Wagner et al).”

6) I cannot find a text reference for Figure S4G.

We have added this now. The figure shows that scRNA-seq and CyTOF data agree as to the proportion of PD-1 high T cells.

7) Line 246: “HLA-ABC” is not a protein. It refers to HLA-A, HLA-B, and HLA-C, which are all MHC class I proteins.

We have rephrased this.

8) Pseudobulk comparison of 9 selected genes in T cells and NK cells does not seem statistically appropriate. Given the genome-wide nature of the dataset and global perspective of the paper, this analysis should control the genome-wide false discovery rate. The effect sizes can still be discussed, but I would recommend not using a cherry-picked list to assess statistical significance.

The list was indeed selected, but we did this based on the literature, to represent a function (cytotoxicity) we were interested to examine. The full genome-wide comparison, including FDR correction, is shown in Figure 2.

9) Call to Figure S4F in line 309 appears wrong.

We have corrected this.

Reviewer #4, expertise in T cell exhaustion (Remarks to the Author):

In this study, Tietscher et al. attempt to compare transcriptional signatures of exhausted T cells in luminal breast cancer. The focus of the study is of interest to the field, however, we question the novelty of this manuscript as it stands. We would recommend further work to extend the mechanisms and pathways highlighted in this work.

1) First and foremost, much of the most significant differences shown between IE1 and IE2 were circular. The initial classification of tumors was described as follows: "samples contained exhausted T cells (i.e., PD-1^{high}/CTLA-4^{high}/CD38^{high} T cells); we annotated these as immune environment 1 (IE1). The other half mainly contained T cells that did not express exhaustion markers and were annotated as immune environment 2 (IE2)". Therefore, the finding that there are more exhaustion-associated markers such as BATF/TOX and more exhausted/proliferative cells in IE1 is unsurprising and almost definitional. Similarly, the finding in Fig. 4 that TAMs and cDC2s (which are associated with wound-healing) are enriched in immune-suppressed tumors are not surprising.

We agree that other recent studies have shown an association between exhausted and proliferative T cells, and between exhausted and tumor-reactive T cells, and we cite these studies in our manuscript. Whether these studies can be called "definitional" is a matter of opinion however. We think our observations add substantially to the body of work on these tumor microenvironments, especially in regard to luminal breast tumors.

In regard to TAMs, we do not report a general association of TAMs with the IE2 environment, but of a specific TAM cluster (TAM-2), while a number of other TAM clusters show a tendency towards enrichment in the IE1 environment.

Finally, we are not aware of previous reports of an association of cDC2s with tumors enriched in non-exhausted T cells, as we report here.

2) The identification of TLS differences between IE1 and IE2 tumors is potentially interesting, but this part of the manuscript needs to be clarified and strengthened. Would the analyses in Fig6A - D look different between IE1 and IE2 if the tumors were separated into TLS and non-TLS areas?

This is an interesting question. We assume, by "TLS and non-TLS areas", the referee means "TLS-containing and non-TLS containing images" and not some more fine-grained spatial definition.

We have compared TLS and non-TLS images for the features reported in Figures 6A, C and D (not applicable for Figure 6B, which shows example images). We repeated the analysis in Figure 6A with TLS images only, this is shown in Figure S10F, and was included in the initial submission. The relevant text in the manuscript reads “Neighbourhood analysis on images containing mature or immature TLS further revealed an enrichment of direct PD-1^{high} T cell-migDC interactions in these images compared with the full dataset (Fig. S10F, compare to Figure 6A).” The two panels are reproduced here for convenience (Figure R14).

Figure R14. Neighbourhood analysis in all images (left) and in TLS-containing images (right). The heat maps show significant pairwise cell type interaction or avoidance on individual images. Square color indicates percentage of images with significant interaction or avoidance ($P < 0.01$), corrected for cell type frequency. Reproduced from Figure 6A (left) and Figure S10F (right).

We repeated the analysis in Figure 6C with TLS images and non-TLS images, and see no difference in the results (Figure R15). We have retained the original Figure 6C in the manuscript.

Figure R15: Paired box plots comparing the percentage of $PD-1^{high}$ T cells versus $PD-1^{low}$ T cells that have at least one migDC as a direct neighbor, when considering only TLS images (left) or only non-TLS images (right). Each pair of dots represents a separate sample. A paired Wilcoxon rank sum test was used for statistical analysis.

We repeated the analysis in Figure 6D separately for TLS and non-TLS images, and again do not find significant differences beyond those already reported. We have revised Figure 6D to now include non-TLS images, reproduced here (Figure R16).

Figure R16. Heatmap displaying the average relative proportion of each cell type among the 10 nearest neighbours for each T cell subtype across all non-TLS images (left) and all TLS-images (both mature and immature, right). Reproduced from revised Figure 6D.

Were migDCs, which seemed to have equal numbers in IE1 vs IE2 tumors by scRNAseq, differentially abundant by spatial analysis?

Yes, we see spatial patterns of migDCs. They are enriched in TLS-containing images (mature and immature) in both IE1 and IE2 tumors; this can be seen in the top left panel in Figure S10D, reproduced here (Figure R17). However, we did not analyse differential abundance of migDCs in IE1 and IE2 in the IMC data, as the need for ROI selection makes these data less unbiased and comprehensive than the scRNA-Seq.

Figure R17. Boxplots comparing the frequency of all cell subtypes for non-TLS, immature TLS, and mature TLS images. The analysis is based on the Protein Panel data and Wilcoxon rank sum test was used for statistical analysis. Reproduced from revised Figure S10D.

Since CXCL13 "did not differ significantly between images with immature versus mature TLS", are there other possibilities in the cytokine profiles that might be candidates for later experimental validation?

We looked into this and found that the proportion of cells expressing CCL5 and CXCL9 was significantly higher in images containing immature TLS compared to mature TLS regions (new Fig S10G, reproduced here as Figure R18). These could be candidates for later experimental validation.

Figure R18. Boxplots comparing the proportion of cells expressing the indicated cytokines in images containing immature versus mature TLS. Wilcoxon rank sum test was used for statistical analysis. Reproduced from revised Figure S10G.

Finally, do the authors see any evidence of direct tumor cell/T cell interactions and if so, which T cell types are found most proximal to the tumor cells?

The neighborhood analysis in the manuscript showed that T cells (like other immune cells) border tumor cells less often than would be expected for a random distribution. However, direct tumor cell/T cell interaction does occur, and IMC data show that an average of 25% of T cells in IE1 samples and 10% of T cells in IE2 directly border a tumor cell (Figure R19). Within individual samples, we see no evidence that any specific T cell type is more proximal to the tumor cells than others; this might be due to the small sample size and the limited number of T cell markers in the IMC panel.

Fig R19: Proportion of T cells directly bordering a tumor cell in IE1 versus IE2. Each dot represents a patient sample. Only non-TLS images were included in the analysis.

3) The goal of identifying a program of exhaustion in luminal breast cancer is laudable, however it is not clear how these findings compare to other single-cell profiling studies in TNBC or other breast cancer types. It is critical to compare these data to other published datasets, both to confirm the robustness of these findings in other cohorts and also to understand which of these findings might be context-dependent vs. represent pan-breast-cancer biology. Some possible refs that are already cited by the authors and are worth comparing to:

Bassez, A. et al. A single-cell map of intratumoral changes during anti-PD1 treatment of patients with breast cancer. *Nature Medicine* (Springer US, 2021).
doi:10.1038/s41591-021-01323-8.

Azizi, E. et al. Single-Cell Map of Diverse Immune Phenotypes in the Breast Tumor Microenvironment. *Cell* 174, 1–16 (2018).

Wagner, J. et al. A Single-Cell Atlas of the Tumor and Immune Ecosystem of Human Breast Cancer. *Cell* 177, 1–16 (2019).

Qian, J. et al. A pan-cancer blueprint of the heterogeneous tumor microenvironment revealed by single-cell profiling. *Cell Res.* 30, 745–762 (2020).

We have integrated our data set with that of Bassez et al (cohort 1) and that of Qian et al. These are the two most recent and largest scRNA-Seq breast cancer data sets available. Briefly, cohort 1 of Bassez et al includes 29 primary breast cancer patients of all subtypes, treated with anti-PD1 before surgery. Tumor samples were taken pre-treatment and on treatment and included 175,942 cells total; we included only pre-treatment samples in our analysis. The cohort of Qian et al includes 14 treatment-naïve breast cancer patients. Primary tumor samples included 44,000 cells of all cell types, of which 14,400 were T cells.

Wagner et al describes a mass cytometry study from our own lab on the same cohort, and was used to classify samples into the two immune environments we have studied here; we have therefore not integrated the data sets.

We have used Seurat for integration, focusing on either immune cells or tumor cells separately; including all cells required too much computational power. We then re-clustered the integrated data set either for broader cell types or more detailed subtypes. There is a good correspondence between the immune cell phenotypes/clusters found in each dataset (Figure R20A-C).

To test whether our major conclusions extend to these other datasets as well, we did the following analyses. First, we tested our observation that tumor cells in IE1 environments have higher MHC-1 expression. We observe that, in all three data sets, there is a correlation between tumor cell MHC-1 expression (HLA-A, B, and C expression) and the proportion of CD8_exhausted or Tfh cells out of all T&NK cell (Figure R20D, left panel).

Second, we tested our observation that CD8 cells expressing exhaustion-like markers also express proliferation markers. We observe that, in all three data sets, there is a correlation between the proportion of CD8_exhausted T cells and of proliferating T cells, out of all T&NK cells (Figure R20D, right panel).

Third, we tested our observations that CXCL13, CSF1 and Granzyme B are differentially expressed in T&NK cells of IE1 and IE2. For this analysis, we used our dataset (Tietscher) to define a threshold we could use to classify the samples in Bassez et al and Qian et al into IE1 and IE2; this threshold was based on the proportion of CD8_exhausted or Tfh cells out of all T&NK cells (Figure R19E). We then used this classification to probe the expression levels of CXCL13, CSF1 and Granzyme B between these sample types (Figure R20F). The CXCL13 and Granzyme B result could be reproduced in both additional data sets. The CSF-1 data showed the same trend but did not reach statistical significance in the Qian dataset, perhaps reflecting the small numbers of cells that express this cytokine.

*We thank the referee for recommending this analysis, which has strengthened several of our conclusions. We now mention these analyses in the revised manuscript and show the data in Figure **S14**.*

Figure Legend:

- A)** UMAP of merged Bassez, Qian and Tietscher datasets, colored by original cell type annotation and dataset. Only immune cells were included and 20,000 cells per dataset were randomly subset.
- B)** Same UMAP as in A, colored by newly generated and annotated clusters.
- C)** Stacked barplot showing the proportion of cells from each dataset in each cluster.
- D)** Left: Scatterplot of the proportion of CD8_exhausted and Tfh cells (out of all T&NK cells) versus the mean epithelial HLA-ABC expression for each patient sample, colored by dataset. Right: Scatterplot of the proportion of CD8_exhausted (out of all T&NK cells) versus the proportion of T_proliferating (out of all T&NK cells) for each patient sample, colored by dataset. Spearman correlation coefficient and p value are indicated for each dataset separately.
- E)** Same scatterplot as in D (left), colored by original IE (Tietscher dataset). The dashed line indicates the cutoff that was selected to assign IE class to the Bassez and Qian samples.
- F)** Comparison of the CSF1, CXCL13 and GZMB expression levels of T&NK cells from samples that were assigned to IE1 versus samples assigned to IE2, separated by dataset. For the Tietscher dataset, the original IE class was used. Wilcoxon testing was used to calculate p-values.

Reviewers' Comments:

Reviewer #1:

Remarks to the Author:

The authors have addressed my concerns. The manuscript is greatly improved.

Reviewer #2:

Remarks to the Author:

I thank the authors for clarification and revision of the manuscript.

Reviewer #3:

Remarks to the Author:

The authors have addressed all of my concerns, and I believe that the manuscript has improved with the modifications made in response to all reviewer comments. While the relatively small number of patients and lack of functional assays preclude conclusive translational insights, the comprehensive exploration of different luminal breast cancer immune environments is methodologically sound and makes an important contribution to the field.

Reviewer #4:

Remarks to the Author:

The authors have addressed the points raised by the reviewers adequately to strengthen the claims in this manuscript, and I would support publication.